# An integrin/MFG-E8 shuttle loads HIV-1 viral-like particles onto follicular dendritic cells in mouse lymph node

Chung Park, John H Kehrl*

B-cell Molecular Immunology Section, Laboratory of Immunoregulation, National Institutes of Allergy and Infectious Diseases, National Institutes of Health, Bethesda, United States

**Abstract** During human immunodeficiency virus-1 (HIV-1) infection lymphoid organ follicular dendritic cells (FDCs) serve as a reservoir for infectious virus and an obstacle to curative therapies. Here, we identify a subset of lymphoid organ sinus lining macrophage (SMs) that provide a cell-cell contact portal, which facilitates the uptake of HIV-1 viral-like particles (VLPs) by FDCs and B cells in mouse lymph node. Central for portal function is the bridging glycoprotein MFG-E8. Using a phosphatidylserine binding domain and an RGD motif, MFG-E8 helps target HIV-1 VLPs to αv integrin bearing SMs. Lack of MFG-E8 or integrin blockade severely limits HIV-1 VLP spread onto FDC networks. Direct SM-FDC virion transfer also depends upon short-lived FDC network abutment, likely triggered by SCSM antigen uptake. This provides a mechanism for rapid FDC loading broadening the opportunity for rare, antigen reactive follicular B cells to acquire antigen, and a means for HIV virions to accumulate on the FDC network.

## Introduction

Afferent lymphatics deliver local infectious agents and subcutaneously administered antigens via the lymph to nearby lymph nodes (LNs) (*Friedlaender and Baer, 1972*). As the lymph flows through the subcapsular sinus, macrophages that line the sinus floor (SCSMs) can actively uptake antigenic material (*Carrasco and Batista, 2007*; *Fossum, 1980*; *Nossal et al., 1965*). For example, locally injected ultraviolet-light inactivated vesicular stomatitis virus (VSV) rapidly appeared in the lymph and accumulated at discrete sites on SCSMs. While the mechanism responsible for VSV capture and accumulation on the SCSM was not identified, it did not depend upon natural antibody and complement C3 fixation as no defect in SCSM VSV retention occurred in animals that lacked C3. Furthermore, the capture mechanism was not unique to VSV as SCSMs also retained locally injected adenovirus and vaccinia virus as well as HIV-1 and murine leukemia virus like particles (VLPs) (*Junt et al., 2007*). Implicating the SCSM scavenger receptor CD169/sialoadhesin in HIV-1 and MLV-VLPs uptake, antibodies directed against CD169 reduced VLP capture. A known CD169 target sialyllactose is found on gangliosides embedded in the HIV-1 and MLV lipid membranes (*Sewald et al., 2015*). Besides CD169 SCSM likely employ multiple other receptors to capture material flowing in the lymph (*Gray and Cyster, 2012*).

Once bound to an SCSM, a virus such as VSV rapidly translocate along the SCSM membrane penetrating the SCS floor. SCSM membrane processes bearing virus project into the underlying B cell follicle. This allows migrating virus-specific B cells access to the viral particles. In VSV infection, SCSM macrophage depletion compromised viral retention, worsened viremia, and impaired B cell responses (*Junt et al., 2007*). Somewhat analogously, captured murine leukemia virus VLPs translocated along the SCSM membrane becoming available to B1 B cells, a target of MLV infection. Once infected the B1 cells migrated to other lymph nodes to spread the infection (*Sewald et al., 2015*).

*For correspondence:
jkehrl@niaid.nih.gov

**Competing interests:** The authors declare that no competing interests exist.

Neither the VSV or the MLV-VLP studies identified the mechanism that transports viral particles across the SCS floor facilitating their delivery to B cells in the lymph node follicle. Furthermore, neither study examined the transfer of the virions or VLPs to the underlying FDC network. FDCs are critical for humoral immunity as they serve as a long-term repository for unprocessed antigens and form a framework for the germinal center and B cell memory responses (*Mandels et al., 1980*).

In patients infected with HIV-1 the ability of FDCs to retain HIV-1 virion for prolonged durations poses a significant challenge to the eradication of the infection. The FDC HIV-1-reservoir contributes to low level viremia. It also provides a potential source of infectious virus for the viral resurgence that typically occurs following treatment interruption (*Heath et al., 1995*; *Schacker et al., 2000*; *Smith et al., 2001*). Important for FDC retention of virions is the complement receptor 2 (CR2, CD21). FDCs prominently express CR2, which binds the complement components C3d and iC3b. FDCs also express a related protein termed CR1 (CD35), which binds C3b, iC3b, C4b, and iC4b (*Carroll, 1998*). Virions become associated with complement components due to their recognition by natural and specific antibodies that fix complement. This allows FDC complement receptors to retain the HIV-1 virions as immune complexes (IC) on lymphoid organ FDC networks (*Armstrong, 1984*).

Antibodies that recognize CD21/CD35 are used to identify FDCs in tissue sections. While follicular B cells also express CR2, they can be distinguished from FDCs based on their morphology and lower CR2 expression level. Another marker used to recognize FDCs in tissue sections is FDC-M1 (*Haley et al., 1995*). The FDC-M1 antibody binds MFG-E8, a bridging molecule that possesses a phosphatidylserine (PS) binding domain and a second domain that contains an RGD motif (*Kranich et al., 2008*). Human and mouse MFG-E8 have two PS binding domains termed C1 and C2. Human MFG-E8 has a single RGD motif binding domain while the mouse has a short and an extended version, which possesses a solitary, or two RGD motif binding domains, respectively (*Hanayama et al., 2002*). In lymphoid organs, MFG-E8 binds PS bearing apoptotic cells delivering them to αvβ3 integrin bearing tingible body macrophages (TBM). Engagement of αvβ3 integrins promotes the phagocytosis and eventual destruction of the apoptotic cell by the macrophages (*Asano et al., 2004*). Besides, binding apoptotic cells, MFG-E8 and related proteins can bind PS present in viral envelopes (*Morizono and Chen, 2014*). As HIV-1 viral particles bud from the host cell, the viral envelope can selectively acquire proteins and lipids from host membranes. While PS is normally located on the plasma membrane inner leaflet during viral budding the PS becomes exposed on the HIV-1 envelope potentially serving as a target for MFG-E8 and related PS binding proteins (*Morizono and Chen, 2014*; *Wu and Shroff, 2018*). Because MFG-E8 is widely expressed and found at substantial concentrations in blood and potentially the lymph (*Kishi et al., 2017*; *Yamaguchi et al., 2008*); most HIV-1 virions likely bear MFG-E8. This would target HIV-1 virions to monocytes and macrophages that express αvβ3/5 integrins. This would also provide another mechanism in addition to CD169 for SCSMs to capture HIV-1 virions from the lymph.

In this study, we examined the delivery of HIV-1 VLPs to subcapsular sinus macrophages and the underlying FDC network. In addition, we assessed the uptake of HIV-1 VLPs by naïve lymph node B cells and by gp120- specific transgenic B cells expressing b12 heavy and light chains. During these studies, we found that HIV-1 VLPs accumulated at distinct sites on SCSMs, which co-localized with MFG-E8 and αvβ3 integrins. Integrin blockade markedly decreased HIV-1 VLP uptake and the appearance of the VLPs on the FDC network. Our results indicate that SCSMs can capture MFG-E8 bound material including HIV-1 VLPs from the lymph, and that entrance into an MFG-E8 enriched compartment facilitates the transfer of this material to the underlying FDC network. Finally, we show a more dynamic FDC network than previously appreciated, which can directly contact the overlying SCSMs.

## Results

### HIV-1 and murine leukemia virus (MLV) VLPs accumulate on discrete site on SMs enriched for MFG-E8 prior to translocation to the FDC network

The HIV-1 envelope protein gp120 injected near the inguinal LN accumulated on SCSMs located between the LN follicles (*Park et al., 2015*). To check the localization of HIV-VLPs following

subcutaneous injection near the inguinal LN, we used intravital 2-photon imaging to collect large image stacks from the SCS deep into the LN follicle at 6 hr (*Figure 1A*) and 20 hr (*Figure 1B*) post injection. Suggesting that gp120 does not direct HIV-1 virion uptake, the HIV-1 VLPs (NL4.3-GFP) rapidly localized (1 hr) on the SCSMs that overlie the LN follicle rather than those over the interfollicular region (*Figure 1—figure supplement 1A*). To clarify the uptake patterns, we co-injected differentially labeled gp120 and HIV-1 VLPs. Both intravital imaging and thick lymph node section imaging (SIGN-R1 immunostained to delineate the interfollicular channels) confirmed the differential uptake (*Figure 1—figure supplement 1B &C*). At the 6 hr time point some of the VLPs had moved inside the lymph node consistent with early FDC loading (*Figure 1A* and *Video 1*). By 20 hr post injection many of the VLPs had transited deep into the B cell follicle, consistent with extensive FDC loading (*Figure 1B* and *Video 1*). To begin to assess the mechanism of the FDC loading, we first asked whether another viral particle behaved similarly to the HIV-1 VLPs. We co-injected Murine leukemia virus (MLV) VLPs and 3 hr later prepared thick LN slices, which we immunostained for CD21/35 and CD169 to outline FDCs and SCSMs, respectively. Following injection of the VLPs, both VLP sets appeared on overlapping sites on the SCSMs (~88% colocalization index, *Figure 1C*). SCSMs have previously been reported to capture these VLPs (*Sewald et al., 2015*). We also noted that both sets of VLPs localized on lymph node FDCs that had directly contacted the overlying SCSMs. These contacts persisted as they were readily observed at 8 hr post injection (*Figure 1D*).

To confirm the CD21/35 immunostaining, we used another antibody that reacts with FDCs termed FDC-M1. This antibody is known to recognizes MFG-E8 (*Gray et al., 1991*). Again, we co-injected NL4.3-GFP and MLV-RFP and made lymph node sections. Surprisingly, we found that besides identifying the lymph node FDCs, the FDC-M1 antibody precisely delineated the SCSM membrane sites where the VLPs had localized (*Figure 1E*). This is more evident in zoomed images. The FDC-M1 immunostaining often encased the VLPs, and when not confluent appeared vesicular. These FDC-M1 SCSM defined sites extended into the underlying follicle directly abutting FDCs processes. Intensity mapping of the three signals on SCSMs shows the strong overlap between MFG-E8 and the two VLPs (*Figure 1F*, far right). 3D-reconstruction of thick section confocal images also shows NL4.3-GFP overlying the SCSM MFG-E8$^+$ sites, which we termed the MFG-E8$^+$ compartment (*Video 2*). To confirm the lymph node MFG-E8$^+$ SCSMs captured NL4.3-GFP VLPs we used flow cytometry to analyze lymph node cell preparations. The CD169$^+$ macrophages captured most of the NL4.3-GFP$^+$ signal (*Figure 1—figure supplement 2A*), and when separated based on MFG-E8 expression, the MFG-E8 high subset bound the most (*Figure 1G*, *Figure 1—figure supplement 2B, C*). This was the case after in vivo transfer (3 hr) and following in vitro binding experiments (30 min on ice).

Splenic CD169$^+$ marginal zone macrophages serve a similar function to that of the lymph node SCSM, although they capture pathogens and antigenic material from the blood rather than from the lymph. To check whether blood-borne HIV-1 VLPs would be captured by marginal zone macrophages and transferred to the splenic white pulp FDC networks, we injected NL4-3-GFP into the blood. After 6 hr we collected the spleens of the injected animals and made thick splenic sections for immunohistochemistry (*Figure 1H*). We found a similar scenario as we had noted in the lymph nodes, the accumulation of HIV-1 VLPs on marginal zone macrophages at MFG-E8 rich sites and the appearance of VLPs on the underlying FDC networks. Thus, sinus lining macrophages in contact with the blood or the lymph collect HIV-1 VLPs at sites on the macrophages enriched for MFG-E8. This raised several questions. How does the MFG-E8 become co-localized with the VLPs on the SCSM and the marginal zone macrophages? Are the HIV-1 VLPs coated with MFG-E8? If so, does MFG-E8 target the VLPs to αvβ3 integrins present on the macrophages?

## HIV-1 VLPs bind MFG-E8, which can interact with αvβ3 integrins

First, we tested whether the HIV-1 VLPs could bind MFG-E8. HIV-1 envelopes are known to expose PS (*Aloia et al., 1993*) providing a potential mechanism for MFG-E8 binding. To determine whether MFG-E8 can bind the HIV-1 VLPs, we used stimulated emission depletion (STED) microscopy to examine NL4.3-GFP VLPs incubated with directly labeled recombinant MFG-E8. The images show MFG-E8 encasing the VLPs (*Figure 2A*, *Figure 2—figure supplement 1A–F*). As expected MFG-E8 bound PS, but not phosphatidylcholine (PC) liposomes (*Figure 2A*). Signal intensity analysis showed MFG-E8 surrounding the NL4.3-GFP core (*Figure 2A*, right panels). Due to their strong, condensed GFP signal, we could detect the VLPs by flow cytometry. This provided another method to test

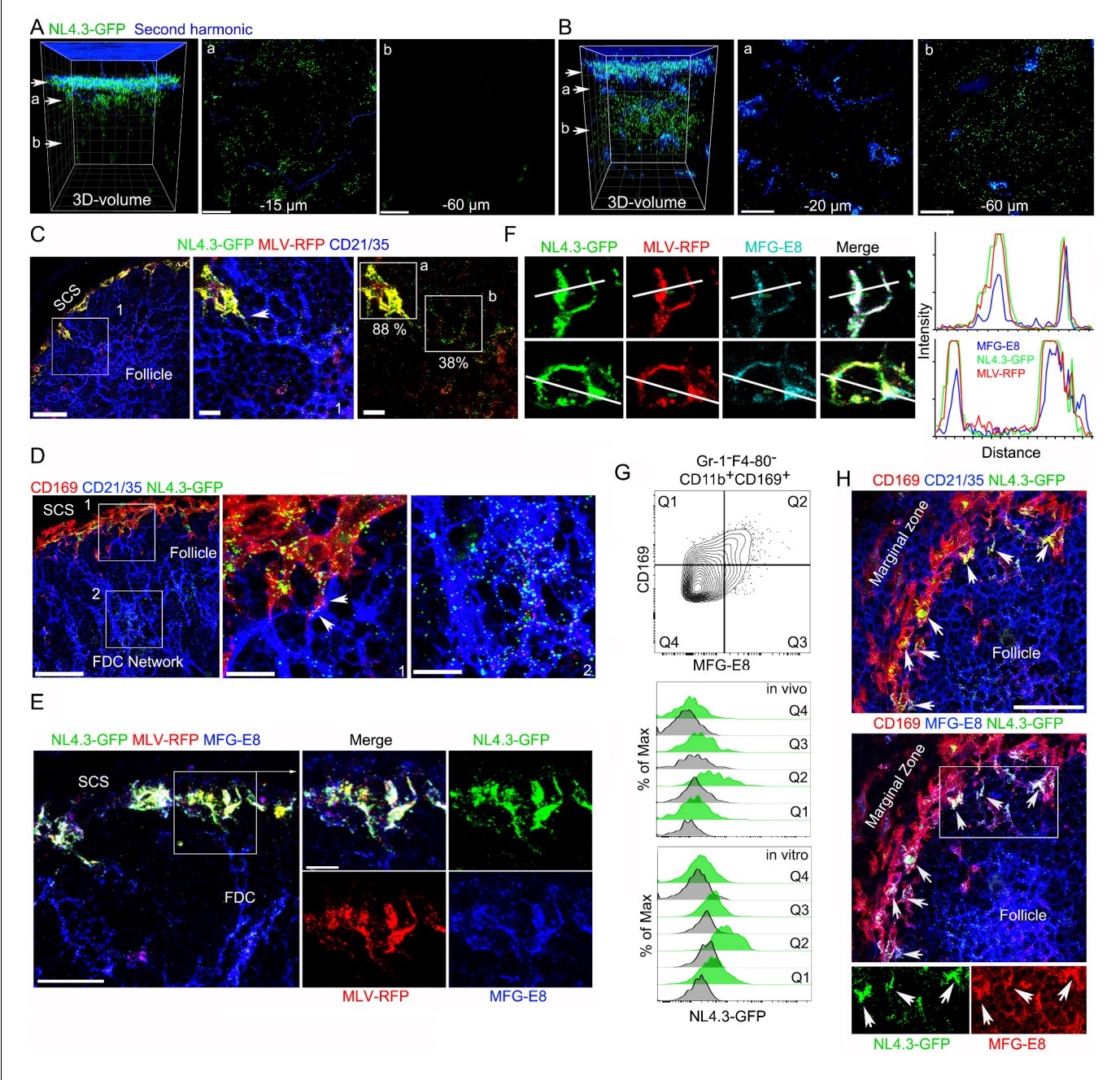

**Figure 1.** VLPs accumulated in MFG-E8+ compartment in SCSM and on the FDC network. (**A, B**) LN TP-LSM thick volume micrographs of NL4.3-GFP (green) 6 hr (**A**) and 20 hr (**B**) post VLPs injection. LN structure visualized by second harmonic (blue) of capsule and sinus floor. In left panel the top arrow indicates the subcapsular sinus floor and arrow a and b the location of the cross-sectional 10 μm volume image shown in panels a and b. Distance from the sinus floor to indicated areas are shown. Grid spacing in volume images is 20 μm. Scale bars 20 μm. (**C**) LN confocal micrographs of NL4.3-GFP and MLV-RFP 3 hr post VLPs injection. Box 1 in first panel zoomed in 2nd and 3rd panels. Arrowhead in 2nd panel indicates a contact point between SCSM and FDC. Colocalization scores of NL4.3-GFP and MLV-RFP in SCSM and FDCs are noted. (**D**) LN confocal micrographs of NL4.3-GFP 8 hr after NL4.3-GFP injection. Boxes in upper panel zoomed in middle and lower panel. Middle panel arrowheads indicate abutment of SCSM and FDC. (**E**) Confocal micrographs of NL4.3-GFP and MLV-RFP in inguinal LN 3 hr post VLPs injection. FDC networks were visualized with FDC-M1 (MFG-E8) antibody. Box in left panel was electronically zoomed. (**F**) Two SCSMs were analyzed for the co-localization of NL4.3-GFP, MLV-RFP and MFG-E8. Region of interest (ROI) in each panel are indicated with white lines. Signal intensity in ROIs were plotted. (**G**) Flow cytometry of LN cells. Gr-1−F4-80−CD11b+CD169+ population from total LN cells was plotted with MFG-E8 versus CD169, which generated Q1, Q2, Q3, and Q4. Histograms show NL4.3-GFP signal (green) and background control (gray) of indicated populations. Upper, LN cells from NL4.3-GFP injected mice. Lower, purified LN cells incubated with NL4.3-GFP. (**H**) Spleen confocal micrographs of NL4.3-GFP 6 hr after intravenous VLP injection. Upper panel, FDC networks

*Figure 1 continued on next page*

*Figure 1 continued*

visualized with CD21/35 and lower panel, FDC networks visualized with MFG-E8 antibody. In bottom panels are shown separated pseudo-colored signals of boxed area in lower panel. NL4.3-GFP (green, left) and MFG-E8 (red, right). Arrows indicate MFG-E8⁺ compartment. Scale bars 30 μm and 10 μm (A, B, E), 50 μm (F).

The online version of this article includes the following figure supplement(s) for figure 1:

**Figure supplement 1.** NL4.3-GFP rapidly localized on SCSMs that overlie the LN follicle rather than those over the interfollicular region.
**Figure supplement 2.** The lymph node MFG-E8⁺ SCSMs capture NL4.3-GFP VLPs.

whether NL4.3-GFP VLPs bound MFG-E8. We confirmed that the VLPs expressed gp120 by incubating them with the gp120-specific antibody VRC01 (*Figure 2B* Histogram). Next, we incubated the VLPs with unlabeled MFG-E8 and detected bound MFG-E8 with an MFG-E8-specific antibody. A low level MFG-E8 antibody signal in the absence of added MFG-E8 suggests that the VLPs may have bound some MFG-E8 during their preparation. However, the addition of recombinant MFG-E8 to the VLPs resulted in a strong shift in the intensity of the MFG-E8 antibody staining (~90% of VLPs were MFG-E8⁺) (*Figure 2B*, *Figure 2—figure supplement 1G*) and directly labeled recombinant MFG-E8 bound HIV-1 VLPs in a dose dependent manner (*Figure 2—figure supplement 1H*). PS containing, but not PC liposomes competed for the VLP MFG-E8 binding (*Figure 2C*). When we injected mice with labeled PS enriched liposomes the lymph node exhibited an SCSM uptake pattern much like NL4.3-GFP, while inter-follicular and medullary SMs predominately gathered injected PC liposomes (*Figure 2—figure supplement 2A–D*). Adding recombinant αvβ3 integrins allowed detection of αvβ3/MFG-E8/NL4.3-GFP complexes (*Figure 2D*) and αvβ3/MFG-E8/PS-liposome complexes (*Figure 2—figure supplement 3A*). The addition of recombinant MFG-E8 to the VLPs resulted in a strong binding to the αvβ3 integrins (~80% of VLPs were αvβ3/MFG-E8 positive) (*Figure 2—figure supplement 3B*). Low level αvβ3 binding to NL4.3-GFP without added MFG-E8 suggests an endogenous αvβ3 integrin binding partner on the VLPs (*Figure 2D*). Together these results indicate that MFG-E8 bound VLPs likely bind to integrin αvβ3 on SCSM.

## MFG-E8 targets HIV-1 VLPs to murine SCSM and human monocytes

If SCSM αvβ3/5 integrins help capture the NL4.3-GFP VLPs, we should be able to visualize integrin/MFG-E8/NL4.3-GFP VLP complexes present on the SCSMs. To assess this possibility, we injected NL4.3-GFP VLPs near the inguinal lymph and prepared thick lymph node sections. Immunostaining the sections with an αv (CD51) antibody visualized αvβ3/5 integrins co-localized with the NL4.3-GFP VLPs in the SCSM MFG-E8⁺ compartment (*Figure 3A*, *Figure 3—figure supplement 1*). 3D-reconstruction of confocal images shows the overlap of MFG-E8, NL4.3-GFP, and αv integrins (*Figure 3B*). Colocalization analysis showed that approximately 85% of NL4.3-GFP colocalized on αv integrin (*Figure 3—figure supplement 2*). These data indicate that the injected VLPs acquire endogenous MFG-E8, which promotes their capture by SCSMs bearing αvβ3 integrins.

Since most blood-borne HIV-1 is likely MFG-E8 bound, the MFG-E8/virion complex may target αvβ3 expressing peripheral blood leukocytes in infected individuals. As expected, human peripheral blood monocytes expressed αvβ3 integrins, while CD4 and CD8 T cells did not (*Figure 3C*), nor did peripheral blood B cells (data not shown). To test mononuclear cell MFG-E8/VLP binding we incubated human peripheral

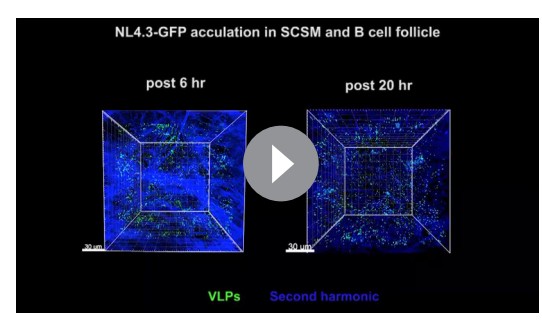

**Video 1.** TP-LSM thick volume images show accumulation of VLPs in SCSM layer and FDC network. Animation shows a volume image of B cell follicle by TP-LSM. The volume image was acquired from inguinal LN of live mouse at 6 hr and 20 hr after VLPs injection. LN structure was visualized by second harmonic (blue) of capsule and sinus floor. Subcapsular floor and follicular cortex are indicated with arrows. Accumulations of NL4.3-GFP (green) is shown in SCSM rich floor at 6 hr post VLP injection and in FDC rich follicular center at 20 hr post VLP injection. Scale bars 30 μm. The volume image spacing is 10 μm.
https://elifesciences.org/articles/47776#video1

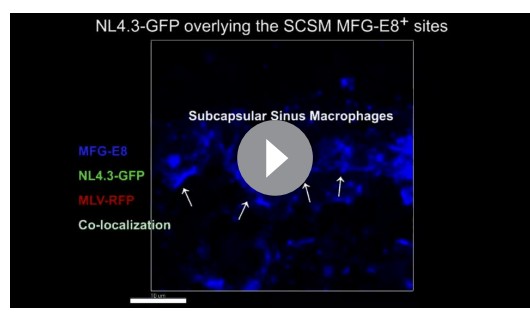

**Video 2.** 3D-reconstruction of thick section images shows NL4.3-GFP overlying the SCSM MFG-E8+ sites, which we termed the MFG-E8+ compartment. Animation shows volume image of thick section confocal image and its 3D-reconstruction. Thick section from inguinal LN collected 3 hr after VLPs injection was stained with FDC-M1 (MFG-E8, blue). NL4.3-GFP (green) and MLV-RFP (red) in MFG-E8+ compartment of SCSMs are indicated with arrows.

https://elifesciences.org/articles/47776#video2

blood mononuclear cells on ice with NL4.3-GFP in the presence or absence of recombinant human MFG-E8. In the absence of added MFG-E8 we observed very low-level VLP binding to some donor's monocytes (*Figure 3D*), however, the addition of MFG-E8 resulted in a substantial increase in NL4.3-GFP binding as approximately 25% of the cells had both an MFG-E8 and a VLP signal (*Figure 3E*). Another donor's monocytes had high levels of constitutive NL4.3-GFP binding, which was enhanced by adding recombinant MFG-E8 (*Figure 3F*). PS containing liposomes, but not PC liposomes competitively inhibited the MFG-E8/VLP binding (*Figure 3F*). Thus, MFG-E8 bound HIV-1 virions likely target human peripheral blood monocytes by binding to αvβ3 integrins.

## Blocking SCSM αv integrins inhibits NL4.3-GFP VLP SCSM capture and transfer to the FDC network, while the lack of MFG-E8 predominately inhibits VLP transfer to the FDC network

To determine whether RGD binding integrins participate in SCSM NL4.3-GFP capture, prior to injecting NL4.3-GFP, we injected a circularized RGD peptide as a competitive inhibitor or an αv blocking antibody (αCD51). Both approaches reduced the SCSM NL4.3-GFP signal, but only by 20–30% (data not shown); however double blockade (RGDyK + αCD51) strongly reduced SSCM uptake (*Figure 4A*, *Figure 4—figure supplement 1A,B*). To visualize SCSMs in the absence of CD169 antibody injection, we employed CX₃CR-1-GFP mice as recipients. Additionally, to provide some insight into the kinetics of the double blockade, we simultaneously injected NL4.3-mCherry VLPs with the blocking agents, thereby impacting the later uptake more than the initial uptake. Isotype control and Fc blocking antibody injected animals exhibited no difference in NL4.3-mCherry uptake (*Figure 4—figure supplement 1C*). The double blockade partially inhibited the initial NL4.3-mCherry binding (below 10 μm) and strongly inhibited the later binding (top 10 μm), (*Figure 4—figure supplement 1D*). LN section imaging confirmed the efficacy of the αCD51 antibody treatment in blocking CD51 (*Figure 4—figure supplement 1E*). Furthermore, the integrin blockade strongly reduced FDC loading (*Figure 4B*, *Figure 4—figure supplement 2A*) and shifted the VLP targeting to the medullary SMs (*Figure 4—figure supplement 2B*, *Figure 4—figure supplement 3A,B*). Of note, the integrin blockade displaced those NL4.3-GFP VLPs captured from the SCSM MFG-E8+ compartment (*Figure 4B*, right panels). Additionally, the double blockade reduced the MFG-E8 localized on SCSMs and decreased the FDC-SCSM contacts (*Figure 4—figure supplement 3C,D*).

Next, we tested the impact of MFG-E8 deficiency on VLP uptake. In contrast to our expectation, the injected NL4.3-GFP still accumulated on the *Mfg-e8⁻/⁻* SCSMs at 20 hr post injection. Yet, the MFG-E8 deficiency markedly impaired FDC NL4.3-GFP loading (*Figure 4C*, *Figure 4—figure supplement 4*). Perhaps due to the poor FDC delivery, the *Mfg-e8⁻/⁻* SCSMs accumulated more VLPs than did the controls (*Figure 4D*). As we noted numerous CD169+ blebs on LN follicle FDCs in both the control and KO mice (*Figure 4—figure supplements 4* and *5*), we normalized the VLP signal to the level of CD169+ blebs. Injecting recombinant MFG-E8 along with NL4.3-GFP into *Mfg-e8⁻/⁻* mice partially reconstituted the MFG-E8 high compartment on the SCSMs and helped transfer some of the VLPs to the FDC network (*Figure 4E* and *Figure 4—figure supplement 6*). In addition, injecting labeled MFG-E8 into the *Mfg-e8⁻/⁻* mice together with NL4.3-GFP allowed visualization of MFG-E8/NL4.3-GFP complexes in the subcapsular sinus and on the SCSMs (*Video 3*). Together this data suggests that SCSM αvβ3/5 integrins can shuttle MFG-E8 bound material to the MFG-E8+ compartment. This compartment is open to lymph node FDCs and B cells (see below). SCSMs that lack functional αv integrins to bind MFG-E8/NL4.3-GFP complexes cannot load the MFG-E8+ compartment or FDCs.

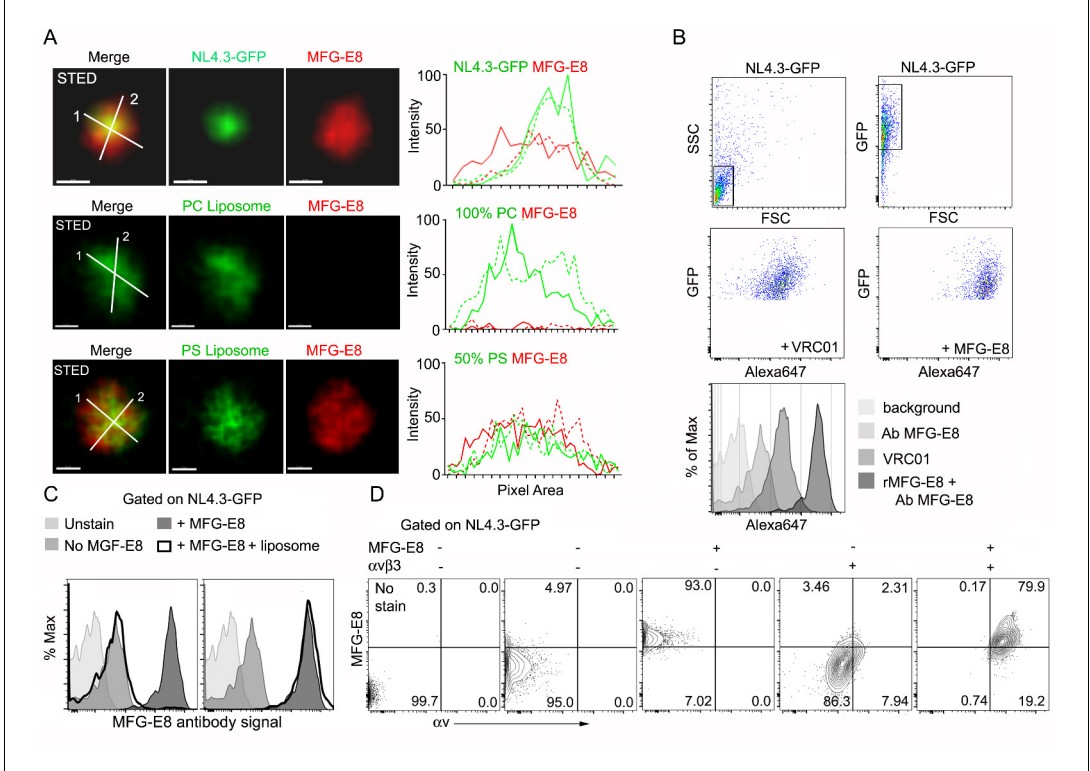

**Figure 2.** HIV-1 VLPs bind MFG-E8, which can interact with αvβ3 integrins. (**A**) STED microscopy images of HIV-1 VLP/MFG-E8 complex (top row), PC-liposome and bound MFG-E8 (middle row), and PS-liposome and bound MFG-E8 (bottom row). Corresponding graphs show signal intensities along axis 1(solid) and axis 2 (dotted line). Scale bars 100 nm. (**B**) Flow cytometry analyzing VRC01 and MFG-E8 binding NL4.3-GFP VLPs. FSC versus SSC shown in 1st panel. 1st panel boxed region used to analyze FSC versus GFP in 2nd panel. GFP versus Alexa 647-VRC01, 3rd panel; and Alexa 647-MFG-E8, 4th panel. NL4.3-GFP VLPs confirmed by VRC01 antibody staining and MFG-E8/NL4.3-GFP complexes detected by MFG-E8 antibody staining and flow cytometry. (**C**) Flow cytometry detects competitive inhibition of MFG-E8 binding NL4.3-GFP by PS-liposome (left), but not PC-liposome (right). (**D**) Flow cytometry detects MFG-E8/αvβ3 integrin/NL4.3 GFP complexes. Binding of recombinant proteins to NL4.3-GFP detected by αv and MFG-E8 antibodies.

The online version of this article includes the following figure supplement(s) for figure 2:

**Figure supplement 1.** MFG-E8 encasing NL4.3-GFP VLPs.

**Figure supplement 2.** PS-liposome target SCSMs, while PC-liposome mainly target medullary and interfollicular channel SMs.

**Figure supplement 3.** MFG-E8 bound HIV-1 VLPs bind to integrin αvβ3.

We remained perplexed by the dissociation between the integrin blockade experiments and the results with the MFG-E8 deficient animals. Perhaps the NL4.3-GFP VLPs acquired low amounts of MFG-E8 during their preparation, which helped promote their capture. The producing cell line HEK293T are routinely cultured in media containing fetal calf serum, a potential MFG-E8 source. To verify we could image an antibody bound to the VLPs, we imaged the HEK293T cells expressing NL4.3-GFP after adding directly labeled VRC01 antibody to the culture. In contrast to negative controls, we found that the majority of the VLPs in the cell culture had bound the VRC01 antibody (*Figure 5A*, *Figure 5—figure supplement 1A & B*). However, VRC01 antibody could not detect delta-Env NL4.3-GFP VLPs or the MLV VLPs (*Figure 5—figure supplement 1C & D*). Next, we added directly labeled antibody against MFG-E8 to a HEK293T culture expressing NL4.3-GFP. Again, many of the NL4.3-GFPs bound the MFG-E8 antibody indicating that had sufficient bound MFG-E8 to allow their detection (*Figure 5B*, *Figure 5—figure supplement 1E* and *Video 4*). Alternatively, the MFG-E8 related protein EDIL-3 (*Hanayama et al., 2004*; *Hidai et al., 1998*) or other envelope integrin binding proteins could compensate for the loss of MFG-E8 in the deficient animals. To test whether EDIL-3 might substitute for MFG-E8 we tested whether recombinant EDIL-3 competed for the binding of MFG-E8 to human monocytes. Adding a 1:1 molar ratio of EDIL-3 to MFG-E8 reduced MFG-E8 binding to human monocytes by approximately 50% (*Figure 5C*,

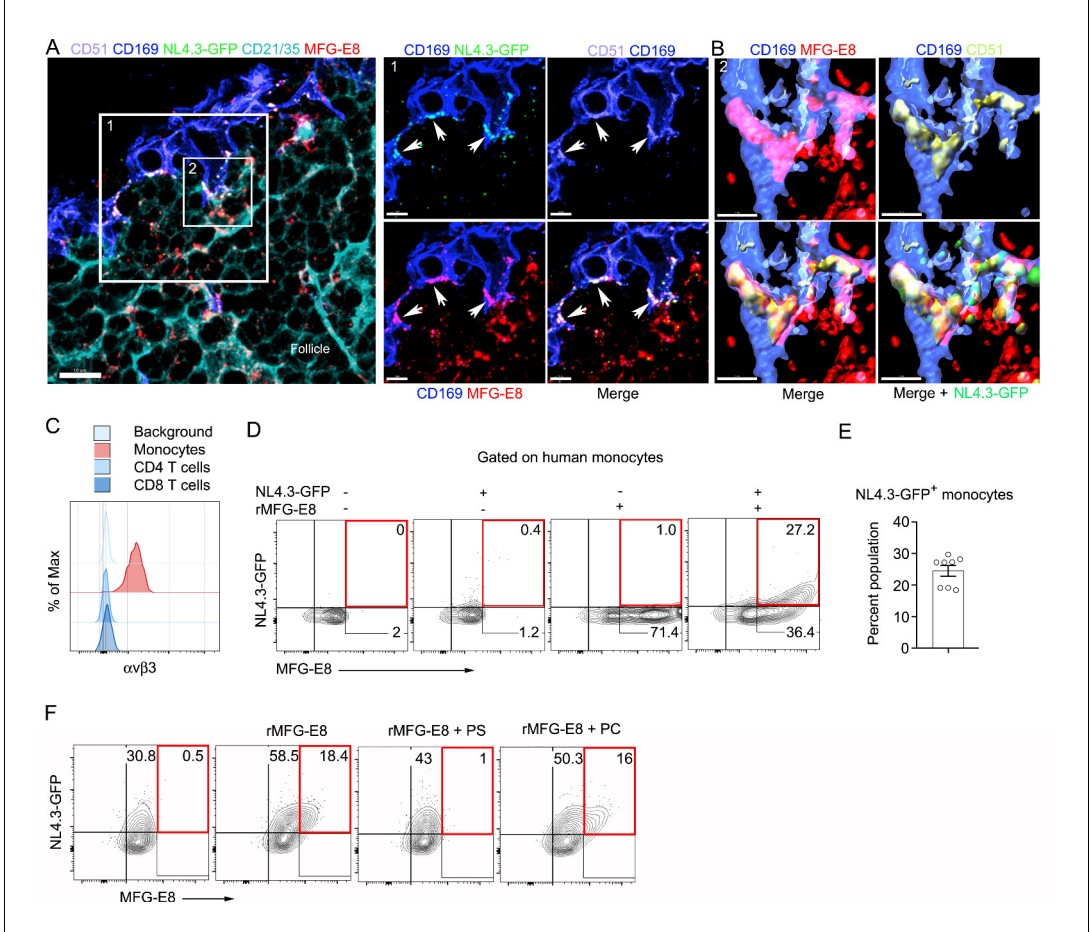

**Figure 3.** MFG-E8 functions as a bridging molecule to target HIV-1 VLPs to murine SCSM and human monocytes. (**A**) LN confocal micrographs of NL4.3-GFP 3 hr after NL4.3-GFP injection. Box 1 zoomed in right panels. Arrow heads indicate overlapping signals. (**B**) Box 2 in (**A**) used for 3D-reconstruction in right panels. Triple signals (CD169, CD51, and MFG-E8) were shown as yellowish compartments in left lower panel. NL4.3-GFP signal was overlapped on the compartment in right lower panel. (**C**) Flow cytometry detects $\alpha v \beta 3$ integrin expression on human monocytes (CD14+ PBMC). (**D**) Addition of recombinant MFG-E8 enhanced NL4.3-GFP binding on human monocytes. Gates were highlighted with red boxes. (**E**) Percent of NL4.3-GFP+ population of human PBMC enhanced by recombinant MFG-E8 pretreatment was plotted. Human PBMC from eight individuals were tested by in vitro binding assay. (**F**) PS-liposome inhibited recombinant MFG-E8 mediated NL4.3-GFP binding on human monocytes. Gates highlighted with red boxes. Scale bars 10 and 5 μm (**A**), and 2 μm (**B**).

The online version of this article includes the following figure supplement(s) for figure 3:

**Figure supplement 1.** Confocal images of isotype control for CD51 staining.

**Figure supplement 2.** NL4.3-GFP VLPs colocalized with αv integrins.

*Figure 5—figure supplement 1F*). These results suggest that HIV-1 virions are likely bound not only by MFG-E8 but also by other PS binding proteins such as EDIL-3.

## Immune complexes co-localize with HIV-1 VLPs on SCSMs and, also trigger the transient abutment of FDCs and SCSMs

Previous studies have shown that SCSM also captures ICs for delivery to FDCs (*Phan et al., 2009*; *Phan et al., 2007*; *Szakal et al., 1983*). To investigate MFG-E8+ compartment involvement in IC uptake and transfer to FDCs, we co-injected preformed phycoerythrin (ICPE) with NL4.3-GFP. Surprisingly, we found a strong overlap between the NL4.3-GFP and ICPE SCSM acquisition sites (*Figure 6A*). Presumably, the SCSMs use different mechanisms for capture, but both types of antigens may move through the MFG-E8+ compartment. Previous studies have not reported direct SCSM-FDCs contact perhaps due to their fleeting nature. Intravital imaging the FDC network beginning 45 min after injection of ICPE showed a dynamic FDC network shift towards SCSMs (*Figure 6B*,

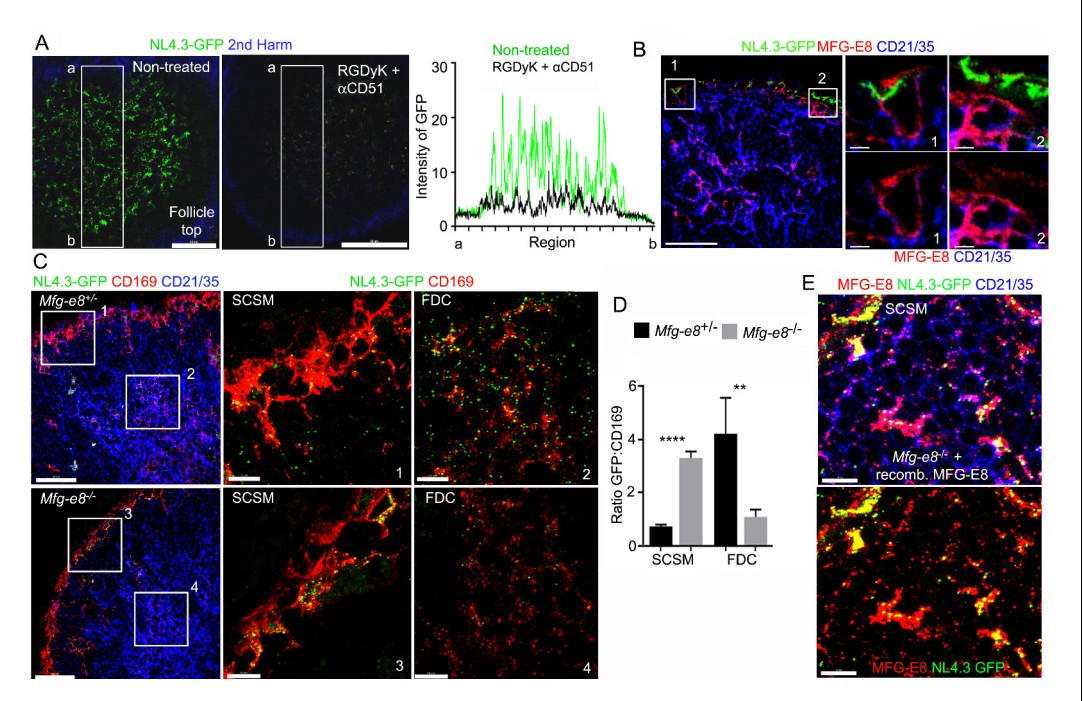

**Figure 4.** Requirement for αv integrins and MFG-E8 to load NL4.3-GFP VLPs on the FDC network. (**A**) LN TP-LSM images from control or from RGDyK (100 µg) + CD51 (10 µg) blocking antibody treated mice captured 80 min after integrin blockade and 20 min after NL4.3-GFP injection. GFP intensity in delineated area shown in graph. (**B**) LN confocal micrographs of NL4.3-GFP from a mouse pretreated with RGDyK + CD51 blocking antibody as same in (**A**). LN collected 1 hr after VLPs injection. SCSMs (in Boxes) zoomed in middle and right panels. (**C**) Confocal micrographs of NL4.3-GFP in *Mfg-e8*[+/−] LN (upper) and *Mfg-e8*[−/−] LN (lower) 20 hr after VLPs injection. Boxes zoomed in middle and right panels. (**D**) Graph of NL4.3-GFP signal overlying SCSM and FDCs normalized to CD169+ deposition signal. Four different sections analyzed and plotted. (**E**) Confocal micrographs of NL4.3-GFP and recombinant MFG-E8 in *Mfg-e8*[−/−] LN 3 hr after VLP injection. MFG-E8 injected 1 hr before NL4.3-GFP. Scale bars 50 µm (**A**), 20 µm and 5 µm (**B**), 50 µm and 10 µm (**C**), and 10 µm (**E**). **; p<0.01, ****; p<0.0001.

The online version of this article includes the following figure supplement(s) for figure 4:

**Figure supplement 1.** Integrin blockade inhibits VLP binding to SCSMs.

**Figure supplement 2.** Integrin blockade shifts VLP targeting to the medullary SMs and strongly reduces FDC loading.

**Figure supplement 3.** Integrin blockade shifts VLP targeting to the medullary SMs and strongly reduces FDC loading.

**Figure supplement 4.** In Mfg-e8[−/−]LNs HIV-1-VLPs accumulate on the SCSMs yet poorly load the FDC network.

**Figure supplement 5.** The deposition of CD169+ blebs on FDCs in the middle of B cell follicle.

**Figure supplement 6.** Recombinant MFG-E8/NL4.3-GFP complexes captured by SCSM and subsequently translocated to FDC network.

*Figure 6—figure supplement 1*, and *Video 5*). The intravital imaging revealed direct contact between SCSM bearing ICPE (or HIV-1 VLPs) and FDCs (*Figure 6C* and *Video 6*). Hypothesizing that the movement of the FDC network depended upon G-protein coupled receptor signaling we inhibited $G\alpha_i$ signaling by injecting pertussis toxin (PTx) near the inguinal LN 2 hr before NL4.3-GFP VLPs. Examining lymph node sections prepared 6 hr later revealed that SCSM had collected the NL4.3-GFP VLPs, but the FDC network had not shifted toward the SSCMs nor had VLPs accumulated on the FDCs (*Figure 6D*, *Figure 6—figure supplement 2*). Furthermore, we tested that PTx impact on FDC movement in real time by intravital imaging (*Video 7*). FDCs and adaptively transferred B cells motility was abolished about 1 hr after PTx treatment. An established abutment of SCSM/FDC by ICPE injection started to dissociate about 2 hr after PTx treatment. These results demonstrate that the antigen established transient abutments of FDCs and SCSMs are temporal and controlled by $G\alpha_i$ signaling.

## gp120-specific B cells extract HIV-1 VLPs from SCSMs

Finally, to test whether the SCSM MFG-E8+ compartment supported NL4.3-GFP transfer to follicular B cells, we adoptively transferred fluorescently labeled wild type and transgenic HIV-1 gp120-

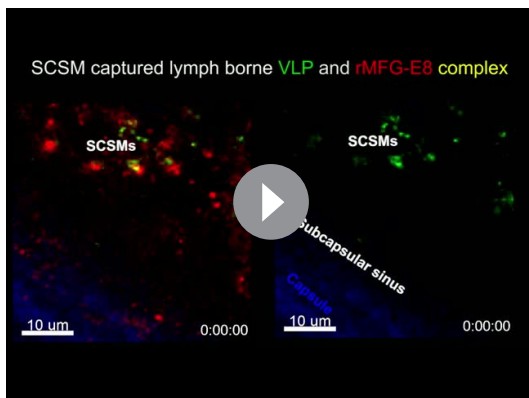

**Video 3.** Visualization of MFG-E8/NL4.3-GFP complexes in the subcapsular sinus and on the SCSMs. TP-LSM live imaging shows complexes (yellow) of NL4.3-GFP (green) and recombinant MFG-E8 (red) in subcapsular sinus of inguinal LN. Recombinant MFG-E8 which was directly labeled with Alexa Fluor 594 was incubated with NL4.3-GFP for 30 min. This mixture was injected into tail base of MFG-E8 KO mouse. Images were acquired from 25 min after injection. Second harmonic signal (blue) delineated capsule of LN. Highlighted circle indicated visible MFG-E8/NL4.3-GFP complex which captured by SCSM in sinus. The right panel video shows signals without MFG-E8 signal from the left video. Second part of video shows that complexes of endogenous MFG-E8/NL4.3-GFP on SCSM. NL4.3-GFP (green) was injected 1 hr before imaging and endogenous MFG-E8 (red) was visualized by antibody injection into tail base (5 μg/mouse) at 30 min before imaging. Arrow indicated a typical MFG-E8/NL4.3-GFP complex. Time counter is hour:minute:second.

https://elifesciences.org/articles/47776#video3

specific b12 B cells (*Ota et al., 2013*) the day prior to injecting NL4.3-GFP VLPs. Lymph node section prepared 3 hr after VLP injection revealed that the transgenic B cells localized at the B-T border, along the inter-follicular channel, and near the SCSMs. In contrast, the transferred wild type B cells tended more towards the center of the follicle, showing little sustained interest in the VLP bearing SCSMs (*Figure 7A* and *Video 8*). Intravital imaging showed b12 B cells extending filopodia to actively probe the NL4.3-GFP bearing SCSMs. Following VLP contact, the B cells reverse direction pulling the VLP now localized on its uropod (*Video 9*) perhaps explaining why B cells carry antigen on their uropod. Zooming in revealed b12 B cells directly contacting the SCSMs bearing NL4.3-GFP VLPs (*Figure 7B*). Reconstruction of the imaging data shows a b12 B cell extending a filopodium to directly contact an MFG-E8$^+$ SCSM bearing NL4.3-GFP VLPs (*Figure 7C* and *Video 10*). We also added b12 B cells to the NL4.3-GFP transfect HEK293T cell cultures along with directly labeled MFG-E8, this allowed us to visualize the transgenic b12 cells capture MFG-E8 coated NL4.3-GFP VLPs (*Figure 7D*) Intravital imaging results indicated that the b12 B cells were attracted to, and in contact with NL4.3-GFP bearing SCSMs, while wild type B cells failed to develop sustained interactions (*Figure 7E* and *Video 8*). We compared the B cells behavior patterns by examining the intravital imaging generated cell tracks. Although the WT and b12 B cell tracks did not have significantly different displacements, the b12 B cells had a reduced mean speed, and turned more frequently (moved less straight).When we focused on those tracks that contacted SCSMs we found

that the b12 B cell interacted with the VLP bearing SCSMs for significantly longer times than did the WT B cells (*Figure 7F*). These results demonstrate that B cells with antigen reactive receptors can use the SCSM MFG-E8$^+$ compartment to retrieve viruses.

## Discussion

Although FDCs are a well-known HIV-1 reservoir, how FDCs initially acquire and preserve HIV-1 remains poorly understood. In this study we used HIV-1 VLPs and mice as a model system to visualize the likely early events in the uptake of HIV-1 virions transiting the lymph and blood by sinus lining macrophages and their subsequent transfer to underlying FDC networks. We identified a mechanism by which an αvβ3 integrin/MFG-E8 shuttle transfers lymph born HIV-1 from SCSM onto FDCs. Furthermore, we suggest that the SCSM MFG-E8 enriched compartment functions as a general antigen transfer portal for B cells and the underlying FDCs. In this paradigm FDCs dynamically react to the delivery of antigen to the overlying SCSMs.

The afferent lymphatics carry lymph from distal sites to the SCS. The lymph flows through the sinus eventually entering the medullary sinuses before exiting the lymph node via the efferent lymphatics. Some lymph-borne material enters the lymph node by exiting the SCS via the conduits. The molecular mass of antigen is a critical factor for how the microarchitecture handles and distributes the antigen (*Gerner et al., 2017*). Lymph-borne viral particles and ICs are too large to enter the

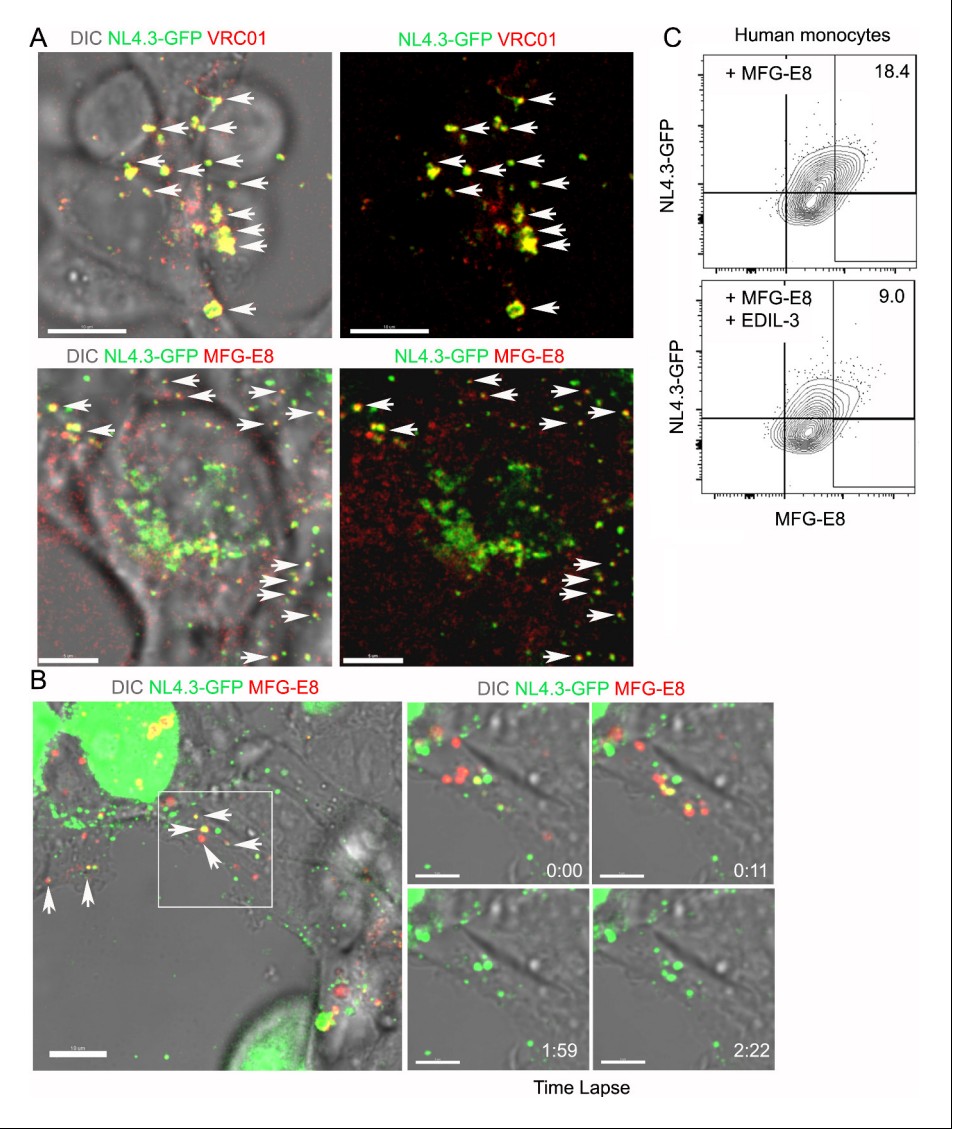

**Figure 5.** MFG-E8/NL4.3-GFP complexes present in transfected HEK293T cell cultures.  (A) Confocal image from live cell imaging of HEK293T cell previously transfected with NL4.3-GFP plasmid. NL4.3-GFP detected by VRC01 staining (upper row). MFG-E8 detected by antibody staining (lower row). Left image includes DIC image. Arrows indicate merged signal. (B) Confocal image from live cell imaging of HEK293T cell previously transfected NL4.3-GFP plasmid. Labeled recombinant MFG-E8 was added to the cells prior to imaging. MFG-E8 bound NL4.3-GFP virions are indicated (arrows). The box in the left panel was electronically zoomed at four different time points. (C) Flow cytometry indicates competitive inhibition of EDIL-3 to MFG-E8 mediated NL4.3-GFP binding on human monocytes. Scale bar 5 μm (A), 10 and 5 μm (B).

The online version of this article includes the following figure supplement(s) for figure 5:

**Figure supplement 1.** Isotype Controls for HEK293T imaging and Edil-3 competition to MFG-E8 binding on NL4.3-GFP VLPs.

conduits. They can be captured by the SCSMs that overlie the lymph node follicle for delivery to underlying cells (*Carrasco and Batista, 2007*; *Fossum, 1980*; *Friedlaender and Baer, 1972*; *Nossal et al., 1965*). Fc and complement receptors assist SCSM in IC capture (*Junt et al., 2007*). CD169 expression helps SCSM capture of HIV-1 and MLV VLPs. Treating mice with an anti-CD169 antibody significantly reduced VLP capture (*Sewald et al., 2015*). The later study found the captured VLPs had accumulated in a virus containing compartment (VCC) contiguous with the SCSM plasma membranes. Consistently, our study found that lymph and blood borne VLPs accumulated on SC

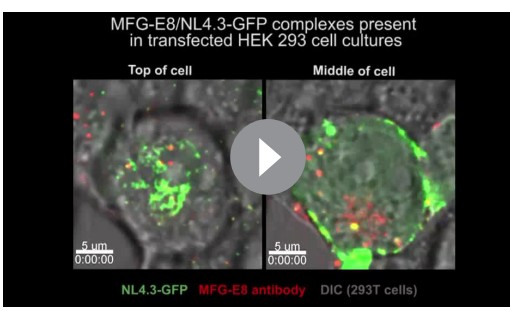

**Video 4.** Visualization of MFG-E8/NL4.3-GFP complexes in transfected HEK293T cell cultures. Time lapse images of HEK293T cells transfected with NL4.3-GFP plasmid 24 hr earlier. A MFG-E8 antibody (red) was added to the culture to visualize NL4.3-GFP/MFG-E8 complexes. The fluorescent images and corresponding DIC image are shown. Two different z-sections are shown, one near the top of the cell and other through the middle of the cell. End of video shows only the fluorescent images. Time counter is hour:minute:second.

https://elifesciences.org/articles/47776#video4

and MZ sinus macrophages, respectively, at sites enriched for MFG-E8. This finding contrasts with our previous report that locally injected gp120 preferentially localized on SIGN-R1[+] macrophages (*Park et al., 2015*) and indicates that the VLP gp120 expression does not mediate their uptake by SCSMs. Confirming that VLP gp120 expression does not affect their SCSM uptake, NL4.3-GFP VLPs lacking gp120 exhibited a similar uptake pattern as did the gp120 expressing VLPs (data not shown). This also agrees with the above study, which noted HIV-1 VLP SCSM acquisition did not depend upon gp120 (*Sewald et al., 2015*). Interestingly, a study using human genetic diversity found that natural CD169 null individuals infected with HIV-1 had no measurable impact on HIV-1 acquisition and progression (*Martinez-Picado et al., 2016*). This argues that early myeloid HIV-1 capture can likely occur by other mechanisms. Together these studies indicate that SCSMs employ a variety of strategies to capture viral particles and that CD169 and MFG-E8 have overlapping roles perhaps functioning at different stages of the capture process.

Ultrastructural studies using electron microscopy to study HIV-1-infected macrophages revealed immature and mature virions seemingly in intracellular vesicles, which resembled late endosomes (LEs) or multivesicular bodies (MVBs) (*Gendelman et al., 1988*; *Wu and Shroff, 2018*). These correspond to the VCC described above and despite their similarity to LEs and MVBs, they are distinct structures that have a complex three-dimensional form, a unique set of protein markers, and several identifying features including a near-neutral pH and the presence of connections to the extracellular milieu. While initially described in response to HIV-1 infection, VCC-like compartments exist in uninfected macrophages (*Tan and Sattentau, 2013*). They often contain other vesicular structures including exosomes (*Wu and Shroff, 2018*). The MFG-E8 enriched compartment described in this study likely correspond to the macrophage VCC. While additional studies are needed to confirm their identity, the SCSM MFG-E8[+] compartment functioned as a niche for VLP accumulation. Our findings also suggest a constitutive, ongoing transport of MFG-E8 bound material from the lymph through this compartment. Supporting this supposition, MFG-E8 bearing SCSMs are evident in the absence of exogenous immune stimulus. The direct transfer of MFG-E8 bearing VLPs to FDCs occurred. The local injection of recombinant MFG-E8 into Mfg-e8[-/-] animals led to the accumulation of MFG-E8 on SCSMs and the eventual appearance of MFG-E8 on the FDC network. MFG-E8 may also be transported in a retrograde fashion from FDCs to the SCSM. Charging the SCSM αvβ3 receptors with MFG-E8 would provide a mechanism to capture HIV-1 virions that lacked bound MFG-E8.

In infected patients, HIV-1 virions accumulate and persist on FDCs for months often as ICs. FDCs endocytose HIV-1 ICs into non-degradative compartments, thereby contributing to the persistence of the virions (*Hart et al., 1991*; *Heesters et al., 2015*; *Smith et al., 2001*; *Smith-Franklin et al., 2002*). An established mechanism to deliver SCSM captured ICs to the underlying central FDC network is via the trafficking of noncognate B cell, which capture the ICs using complement receptors (*Phan et al., 2009*; *Phan et al., 2007*). However, early in the infection prior to IC complex formation other mechanisms likely exist to help transfer antigens from the SCSMs to FDCs. Furthermore, B cell-mediated IC transfer may be inadequate to transfer large particulate antigens such as a virion. In our studies we failed to observe non-cognate B cell uptake and transfer of HIV-1 VLPs to the FDC network (Park, unpublished observation). Rather we consistently observed an early, but transient abutment of the FDC network to the overlying SCSMs, thereby allowing the direct transfer of HIV-1 VLPs to FDCs. Within 24 hr the FDC network had retracted back to its usual location in the center of the follicle accompanied by the captured VLPs. Notably, interfering with the SCSM/FDC abutment

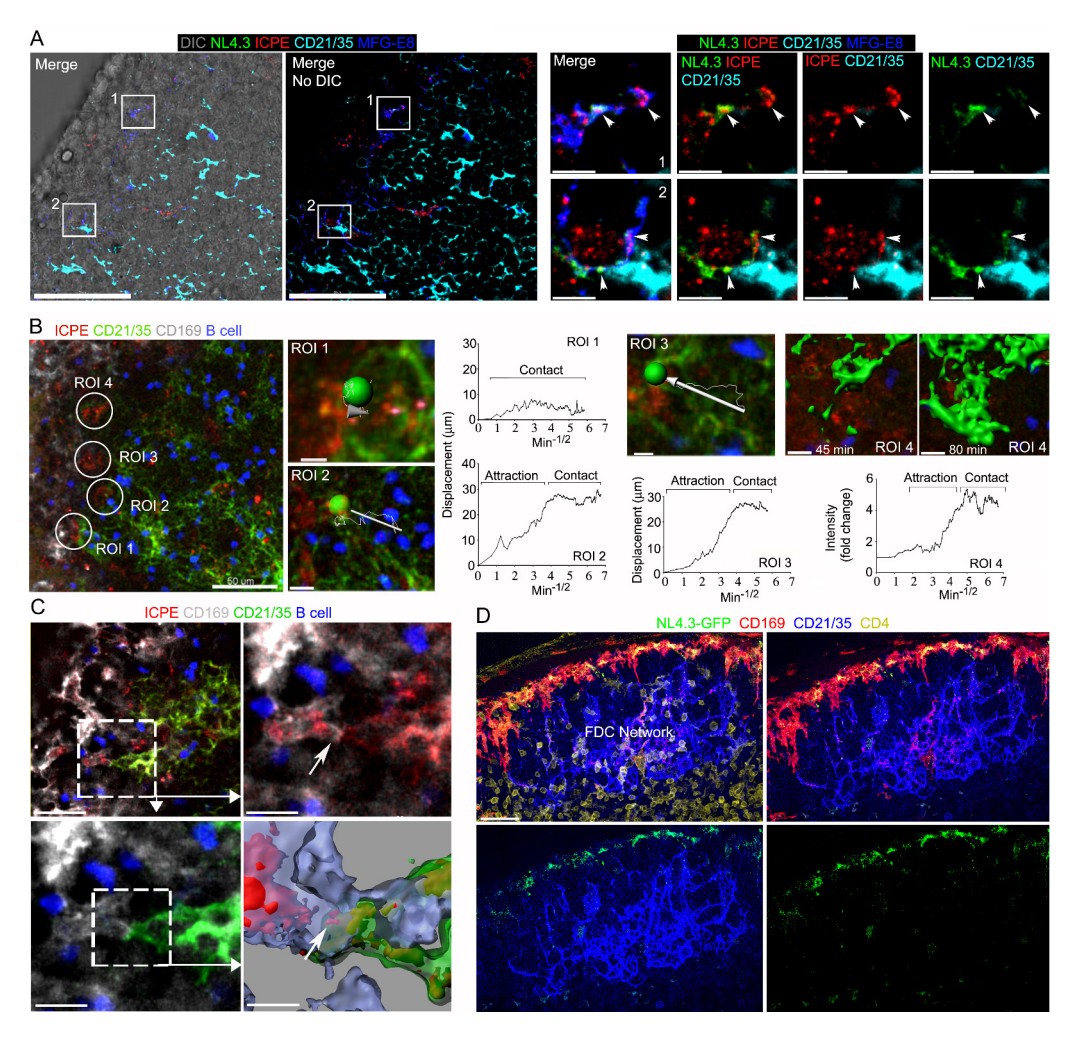

**Figure 6.** Immune complexes co-localize with HIV-1 VLPs on SCSMs and trigger the transient abutment of FDCs and SCSMs. (A) LN confocal micrographs of NL4.3-GFP and ICPE 4 hr after NL4.3-GFP injection. ICPE injected 1.5 hr after NL4.3-GFP. In left panel DIC superimposed image shown. SCSMs in boxes (left and middle panels) zoomed. Arrows indicate NL4.3-GFP, ICPE, and MFG-E8 co-localization. (B) LN TP-LSM image following ICPE injection. In ROI 1–3 arrows show FDC displacement and green balls indicate arbitrary center points of FDC tracked. Graphs show FDC movement relative to SCSM. ROI4 shown as 3D-reconstructed volume of a FDC at indicated times. Graph records time-dependent changes in FDC signal. (C) LN TP-LSM images at SCSM/FDC interface 3 hr after ICPE injection. Dotted white box zoomed in top, right and bottom, left panels. Dotted box reconstructed as 3D-volume, bottom right. Arrows indicate an abutment of SCSM and FDC, which contains ICPE. (D) Confocal images of LN sections showing impact of pertussis toxin on FDC network translocation. Pertussis toxin (500 ng) injected 2 hr before NL4.3-GFP. LN analyzed 6 hr later. Scale bars 50 and 5 µm (A), 50 and 5 µm (B), 30 and 10 µm (C), and 30 µm (D).

The online version of this article includes the following figure supplement(s) for figure 6:

**Figure supplement 1.** A dynamic FDC network shift towards SCSMs.

**Figure supplement 2.** The impact of pertussis toxin on FDC network localization in B cell follicle.

by pre-injecting pertussis toxin severely reduced the rapid VLP loading on the FDCs. The reduced SCSM/FDC abutment by pertussis toxin argues that heterotrimeric Gi protein-mediated signaling contributes to the FDC movement. The most intriguing possibility is a chemoattractant directed movement of the FDC network towards the SCSMs triggered by VLP uptake. However, since the local pertussis toxin injection will affect all the LN cells near the SCS, we cannot conclude that the FDC movement and severe reduction in FDC/VLP loading results solely from reducing the SCSM/FDC abutment. Interestingly, a similar transient FDC-SCSM abutment occurred following the administration of ICs. Identification of the mechanisms underpinning the FDC network movements may provide a means to control the loading of antigen onto FDCs.

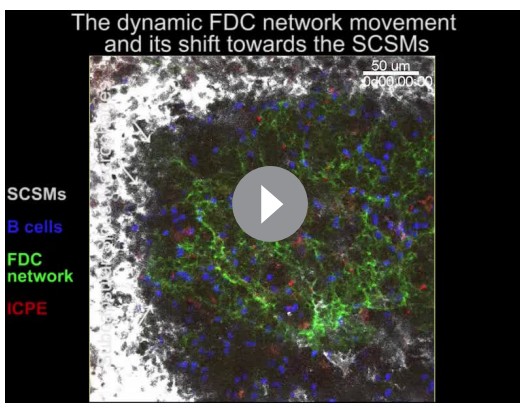

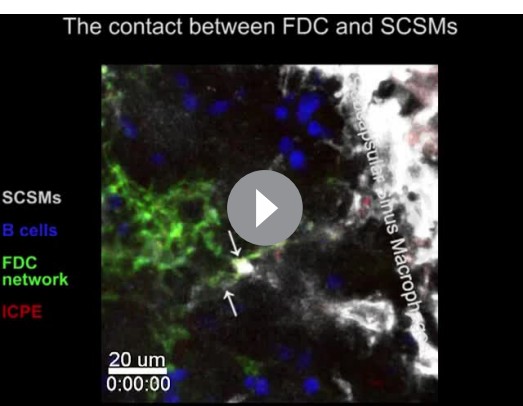

**Video 5.** The dynamic FDC network movement and its shift towards the ICPE bearing SCSMs. TP-LSM live imaging shows the inguinal LN from 45 min to 1 hr 45 min after injection of ICPE (red). SCSMs and FDCs network revealed by injected CD169 (white) and CD21/35 (green) antibodies. B cells (blue) adoptively transferred the previous day outline the follicle. ICPE bearing SCSMs were indicated with arrows. End of two separated video indicate additional examples of FDC network movement. Time counter is hour:minute:second.
https://elifesciences.org/articles/47776#video5

**Video 6.** The direct contact between SCSM bearing ICPE (or HIV-VLPs) and FDCs. TP-LSM live imaging shows the inguinal LN at the SCSM (white)/FDC (green) interface 3 hr after ICPE (red) injection. SCSMs and FDCs network revealed by injected CD169 (white) and CD21/35 (green) antibodies. B cells (blue) adoptively transferred the previous day outline the follicle. The SCSM/FDC interface were indicated with arrows. Second video shows the inguinal LN at NL4.3-GFP bearing SCSM (green)/FDC (red) interface 6 hr after NL4.3-GFP injection. FDCs network revealed by injected CD21/35 (red) antibodies. The third video zoomed in a contact point between SCSM and FDC in which contains HIV-1 VLPs. Arrows indicate VLP transit from SCSM to FDC. The last portion of the video shows the FDC network and VLPs in the B cell follicle at 6 hr after injection. Time counter is hour:minute:second.
https://elifesciences.org/articles/47776#video6

In contrast to our observations, a study using bacteriophage Qβ VLPs demonstrated that non-cognate B cells captured the VLPs and transported them to splenic FDCs. B cell uptake depended upon natural IgM antibody and complement (*Link et al., 2012*). The authors did not examine macrophage uptake and confined their analysis to the spleen 24 hr after VLP injection. Bacteriophage Qβ VLPs are composed of a T = 3 icosahedral lattice of coat proteins assembled with genomic RNA in absent of lipid membrane with a diameter of 28 nm (*Gorzelnik et al., 2016*). These VLPs differ substantially from HIV-1 VLPs as they lack lipid components, are unlikely to bind MFG-E8, and are much smaller (150–200 nm versus 28 nm). These differences likely result in alternative mechanisms of VLP capture and transfer to FDCs. Nevertheless, it would be of interest to determine whether the Qβ VLPs are captured by LN SCSMs and to visualize how they are transported to LN FDCs.

While SCSM use different strategies to capture ICs and the HIV-1 VLPs, both entered the MFG-E8[+] compartment on SCSM. This raised the possibility that this compartment functions as a portal for the transfer of SCSM captured antigens to FDCs and cognate B cells. That MFG-E8 may have a role in the transfer is supported by the poor loading of VLPs on to FDCs in the MFG-E8 deficient mice despite the adequate capture by the SCSMs. Our imaging data clearly shows the direct transfer of both ICs and VLPs from SCSMs to FDCs. An obvious question is how the FDCs acquire the ICs and VLPs from the SCSMs. In the case of ICs FDC complement receptors likely contribute, however, for VLPs the answer is unknown. Still to be determined is whether the VLPs remains MFG-E8 bound and what the role of the αvβ3 integrins play in the MFG-E8[+] compartment. Since αvβ3 integrins support macrophage phagocytosis the SCSM VLPs may internalize the VLPs prior to delivery to the VCC much like exosomes are delivered to the VCC. Furthermore, the binding of the MFG-E8 virion complex to the SCSM αvβ3 integrins may signal the cells via its known role in triggering Rac1 activation (*Mao and Finnemann, 2012*).

Cognate B cells with gp120 reactive B cell antigen receptors (BCRs) readily acquired NL4.3-GFP VLPs directly from the SCSMs. We found b12 B cells extending filopodia to directly contact MFG-

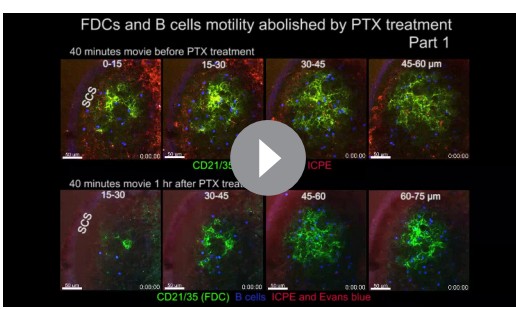

**Video 7.** The impact of pertussis toxin on FDC network and B cell movement in a follicle. In part 1, TP-LSM live imaging movies in upper row show the inguinal LN at the subcapsular sinus (SCS) area for 40 min from 1.5 hr after ICPE (red) injection. FDC (green) contacted and fluctuated around ICPE (red) bearing SCSM. B cells (blue) adoptively transferred the previous day outline a B cell follicle. In the lower row movies, the same subcapsular sinus (SCS) area is shown for 40 min 1 hr after pertussis toxin (PTx, 500 ng) injection. Evans blue (red) was co-injected with PTx to visualize the SCS. Indicated numbers in each movie show the distance from SCSM floor to imaged area. All movies have 15 μm of thickness. In part 2, two comparable videos from part one were enlarged and to show the movement of FDCs and B cells. Arrow in PTx treated video shows loss of transient abutments. Time counter is hour: minute:second.

https://elifesciences.org/articles/47776#video7

E8[+] SCSM bearing NL4.3-GFP VLPs. Eventually, the b12 B cells reversed direction pulling the VLP away from the macrophage. Individual gp120 reactive B cells could acquire and simultaneously transported multiple HIV-1 VLPs on their uropod. This indicates that VLPs in MFG-E8[+] compartment retain gp120 recognizable by the b12 B cell BCR. Our in vitro studies suggest that the b12 B cell BCRs can recognize gp120 even when the VLPs are MGF-E8 bound. Yet whether the binding of MFG-E8 to the HIV-1 virions can shape the viral epitopes available to trigger a humoral response remains an open question. While non-cognate B cells transiently interacted with SCSMs bearing NL4.3-GFP VLPs, they rarely extracted VLPs from them. BCR triggered F-actin remodeling and myosin- and dynein-mediated contractility are known to contribute to the traction force generation needed for antigen extraction (*Wang et al., 2018*). BCR engagement and antigen acquisition likely altered the behavior of the cognate B cells as they slowed and focused their interest on the SCSM bearing VLPs as compared to B cells that lacked gp120 reactive BCRs. Beside engaging the gp120-specific BCRs the MFG-E8 bound VLPs are likely also recognized by αvβ3 integrins expressed by B cells. The engagement of these integrin receptors can be immunosuppressive. The deletion of αv or β3 in B cells enhances the humoral response to antigens containing TLR ligands and to the influenza virus. The enhancement results from the loss of an integrin-mediated negative regulatory signal (*Raso et al., 2018*; *Wang et al., 2014*).

VLPs have been employed as a vaccine strategy in the hope that could advance the elicitation of broadly neutralizing antibodies (bNAbs) against HIV-1 envelope proteins. Multiple reports have demonstrated that HIV-1 VLPs can induce strong antibody responses in small animal models and macaques (*Tong et al., 2014*), (*Buonaguro et al., 2012*). Furthermore, expertise in the generation and usage of VLPs is evolving towards the goal of producing a safe mimic of real HIV-1 virions (*Gonelli et al., 2019*). However, care in the evaluation of HIV-1 VLPs vaccine studies is needed. Variation in packaging cell lines and animal models chosen may result in significant variation and cross-species-specific immune responses. Our study utilized non cell-associated VLPs as a mimic of the early stages of HIV-1 virion dissemination in lymphoid organs. While plasma contains significant levels of cell free virus there is little information on the amount of free virus or cell associated virus in the lymph. Yet as much of the plasma virus is derived from tissue sources, it seems likely that it has transited through the lymph and lymph nodes. Limitations of our study include a reliance on a non-replicating VLP rather than a replicating virus. As such to mimic the transit of infectious virions in the blood and lymph, we had to directly inject the VLPs. Another limitation was the choice of a mouse model. Yet this also provided us with enhanced imaging capability and the ability to use B cells with a defined antigen receptor specificity.

Based on our observations, we propose that MFG-E8 blood levels and lymphoid organ sources of MFG-E8 ensure that most cell-free HIV-1 virus in infected individuals is exposed to MFG-E8. HIV-1 envelope PS provides a means for MFG-E8 binding and MFG-E8 links the virions to αv integrins on host cells. Whether this MFG-E8 opsonization benefits the host or the virus needs further study. MFG-E8 opsonization does promote the delivery of fluid borne virions to SMs resulting in their accumulation in an MFG-E8[+] compartment. This compartment is accessible to antigen-reactive B cells and transiently to FDC processes. Direct SM-FDC virion transfer depends upon a short-lived FDC

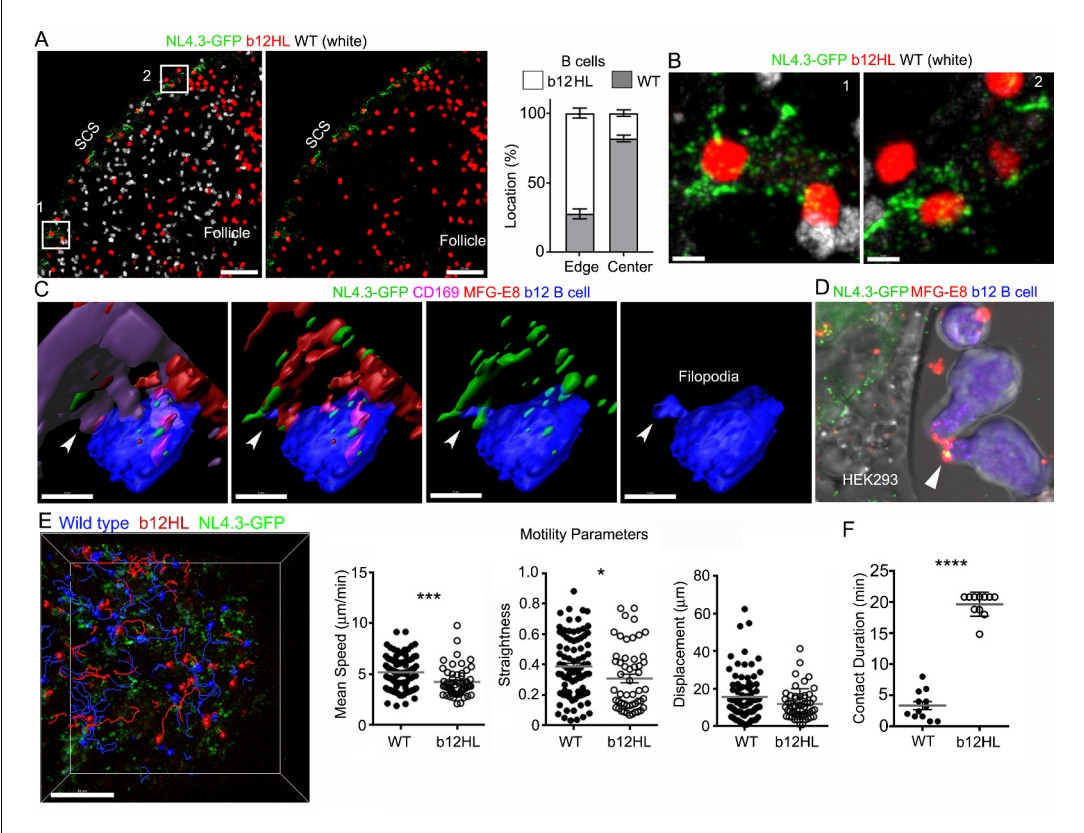

**Figure 7.** gp120-specific B cells extract HIV-1 VLPs from SCSMs. (**A**) LN confocal images of adaptively transferred b12 and wild type B cells 3 hr after NL4.3-GFP injection. B cells transferred day before. Right panel shows only b12 B cells in the follicle. Graphical analysis shows percent ratio of wild type versus b12 B cells in B cell follicle center (Center) and follicular edge (Edge). (**B**) Boxes (1 and 2) in right panel zoomed. (**C**) 3D-reconstructed images of b12 B cell tightly associated with NL4.3-GFP bearing SCSMs. Arrows indicate an MFG-E8+ compartment on SCSM. Final panel shows a b12 B cell filopodia probing MFG-E8+ compartment. (**D**) Confocal image from live cell imaging of HEK293T cell previously transfected with NL4.3-GFP plasmid. Directly labeled recombinant MFG-E8 (red) was added 1 hr prior to imaging. Arrow indicates MFG-E8 binding NL4.3-GFP VLP. (**E**) TP-LSM image overlaid with the tracks of wild type (blue) and b12 B cells (red). B cells were adoptively transferred 1 day before NL4.3-GFP VLP injection. 1 hr after NL4.3-GFP injection intravital images were taken and used to analyzed B cells behavior. (**F**) Motility parameters were calculated and shown. A comparison between contact durations between B cells (WT or b12 B cells) and VLP bearing SCSM are shown (right panel). Scale bars 50 μm (**A**), 5 μm (**B**), 5 μm (**C**), and 50 μm (**E**). *; p<0.1, **; p<0.01, ****; p<0.0001.

network abutment, likely triggered by SCSM antigen uptake. Subsequently, the FDCs network contracts toward the follicle center along with its acquired antigens. This provides a mechanism for rapid FDC loading broadening the opportunity for rare, antigen reactive follicular B cells to acquire antigen, but also a means for HIV-1 virions to accumulate on the FDC network.

## Materials and methods

### Key resources table

| Reagent type (species) or resource | Designation | Source or reference | Identifiers | Additional information |
|---|---|---|---|---|
| Strain, strain background (*Mus musculus*) | C57BL/6J | The Jackson Laboratory | 000664 | |

*Continued on next page*

Continued

| Reagent type (species) or resource | Designation | Source or reference | Identifiers | Additional information |
|---|---|---|---|---|
| Strain, strain background (*Mus musculus*) | Igh[tm1.1(b12)Nemz] | (*Ota et al., 2013*) | Available in The Jackson Laboratory 23065 | A floxed neomycin resistance cassette replaced the DQ52-JH cluster in the Igh locus with HIV antibody b12 variable region coding sequence. Cre-mediated recombination removed the neomycin resistance cassette. |
| Strain, strain background (*Mus musculus*) | Igk[tm1.1(b12)Nemz] | (*Ota et al., 2013*) | Available in The Jackson Laboratory 23065 | A floxed neomycin resistance cassette replace the J cluster in the Igk locus with the HIV antibody b12 L chain VJ element. Cre-mediated recombination removed the neomycin resistance cassette. |
| Strain, strain background (*Mus musculus*) | B6.129P2(Cg)-Cx3cr1[tm1Litt]/J | The Jackson Laboratory | 23065 | $CX_3CR1$-GFP mice |
| Strain, strain background (*Mus musculus*) | Mfg-e8[-/-] (MFG-E8 KO) | (*Neutzner et al., 2007*) | | Milk Fat Globule-EGF Factor 8 (MFG-e8) deficient mice |
| Transfected construct (*Human immunodeficiency virus type 1*) | pCMV NL4-3 Gag-eGFP | (*Guzzo et al., 2017*) | Dr. Walter Mothes (Yale University) | Plasmid for NL4.3-GFP VLPs |
| Transfected construct (*Human immunodeficiency virus type 1*) | pCMV NL4-3 Gag-mCherry | This paper (Materials and methods section) | Dr. John Kehrl (NIAID/NIH) | Plasmid for NL4.3-mCherry VLPs |
| Transfected construct (*Human immunodeficiency virus type 1*) | HIV-1 NL4-3 Gag-iGFP ΔEnv | NIH AIDS Reagent Program | 12455 | Plasmid for delta-Env NL4.3-GFP VLPs |
| Transfected construct (*Friend murine leukemia virus*) | pSV-A-MLV-env | NIH AIDS Reagent Program | 1065 | Plasmid for MLV VLPs |
| Transfected construct (*Friend murine leukemia virus*) | pSV-Ψ-MLV-env⁻ | NIH AIDS Reagent Program | 3422 | Plasmid for MLV VLPs |
| Transfected construct (*Friend murine leukemia virus*) | MLV Gag-RFP | Addgene | 1814 | Plasmid for MLV VLPs |
| Chemical compound, drug | 1,2-dioleoyl-sn-glycero-3-phosphocholine (DOPC) | Avanti Polar Lipids | 850375 | |
| Chemical compound, drug | 1-oleoyl-2-{6-((7-nitro-2–1,3-benzoxadiazol-4-yl)amino)hexanoyl}-sn-glycero-3-phosphocholine (NBD-PC) | Avanti Polar Lipids | 810132 | |
| Chemical compound, drug | 1,2-dioleoyl-sn-glycero-3-phospho-L-serine (DOPS) | Avanti Polar Lipids | 840035 | |

*Continued on next page*

Continued

| Reagent type (species) or resource | Designation | Source or reference | Identifiers | Additional information |
|---|---|---|---|---|
| Chemical compound, drug | 4-Hydroxy-3-nitrophenylacetyl hapten-Phycoerythrin (NP-PE) | Biosearch Technologies | N-5070–1 | |
| Chemical compound, drug | Pertussis toxin (PTX) | Millipore Sigma | 70323-44-3 | |
| Antibody | αv blocking antibody (αCD51) (RMV-7), (Rat monoclonal) | BioLegend | 104108 | in vivo blocking (10 µg) |
| Antibody | Mouse MFG-E8 Antibody (Clone# 340614), (Rat monoclonal) | R and D Systems | MAB2805 | Dilution factor for FACS (1:500), Section (1:1000) |
| Antibody | Human MFG-E8 Antibody (Clone# 278918), (Mouse monoclonal) | R and D Systems | MAB27671 | Dilution factor for FACS (1:500) |
| Antibody | Anti-Mouse Follicular Dendritic Cell (FDC-M1), (Rat monoclonal) | BD Biosciences | 551320 | Dilution factor for section (1:1000) |
| Antibody | anti-human αvβ3 integrin (23C6), (Mouse monoclonal) | BioLegend | 304413 | Dilution factor for FACS (1:500) |
| Antibody | anti-mouse CD21/35 (7E9), (Rat monoclonal) | BioLegend | 123421, 123411, 123409 | BV421, PE, APC, Dilution factor for section (1:1000) |
| Antibody | anti-mouse CD35 (8C12), (Rat monoclonal) | BD Biosciences | 740029 | Dilution factor for section (1:1000) |
| Antibody | anti-mouse CD169 (3D6.112), (Rat monoclonal) | BioLegend | 142403 | Dilution factor for FACS (1:600), Section (1:1500) |
| Antibody | Anti-HIV-1 gp120 Monoclonal (VRC01), (Human monoclonal) | NIH AIDS Reagent Program | 12033 | Dilution factor for FACS (1:500) |
| Peptide, recombinant protein | Recombinant Mouse MFG-E8 Protein | R and D Systems | P21956 | |
| Peptide, recombinant protein | Recombinant Human MFG-E8 Protein | R and D Systems | Q08431 | |
| Peptide, recombinant protein | Recombinant Human EDIL-3 Protein | R and D Systems | 6046-ED-050 | |
| Peptide, recombinant protein | Recombinant Human Integrin alpha V beta 3 Protein | R and D Systems | 2308-VN | |
| Peptide, recombinant protein | Cyclo(-RGDyK) | AnaSpec | AS-61183–5 | |
| Commercial assay or kit | TransIT-293 Transfection Reagent | Mirus Bio LLC | MIR 2704 | |
| Software, algorithm | Imaris software v.9.0.1 64x | Oxford Instruments plc | https://imaris.oxinst.com/ | |
| Software, algorithm | LSA AF Lite software | Leica Microsystem | https://www.leica-microsystems.com/ | |

*Continued*

| Reagent type (species) or resource | Designation | Source or reference | Identifiers | Additional information |
|---|---|---|---|---|
| Software, algorithm | FlowJo | BD Biosciences | https://www.flowjo.com/solutions/flowjo | |
| Software, algorithm | Prism software | GraphPad Software | https://graphpad.com | |

## Mice

C57BL/6 mice were obtained from the Jackson Laboratory. $Igh^{tm1.1(b12)Nemz}$ (b12 Heavy, b12HH) and $Igk^{tm1.1(b12)Nemz}$ (b12 Light, b12LL) mice were obtained from Dr. David Nemazee (*Ota et al., 2013*). C57BL/6(FVB)-$Igk^{tm1.1(b12)Nemz}$ $Igh^{tm1.1(b12)Nemz}$/J (b12 Heavy Light, b12HL) mice were generated by breeding b12HH × b12 LL. M$fg$-$e8^{-/-}$ (MFG-E8 KO) mice were obtained from Dr. Mark Udey (NIAID, NIH) (*Neutzner et al., 2007*). B6.129P2(Cg)-Cx3cr1$^{tm1Litt}$/J (CX$_3$CR-1-GFP) mice were obtained from the Jackson Laboratory. All mice were used in this study were 6–16 weeks of age. Mice were housed under specific-pathogen-free conditions. All the animal experiments and protocols used in the study were approved by the NIAID Animal Care and Use Committee (ACUC) at the National Institutes of Health.

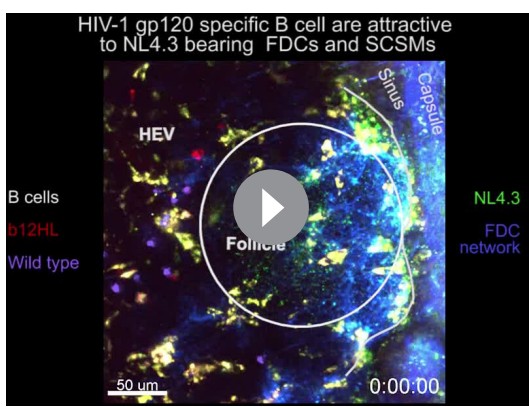

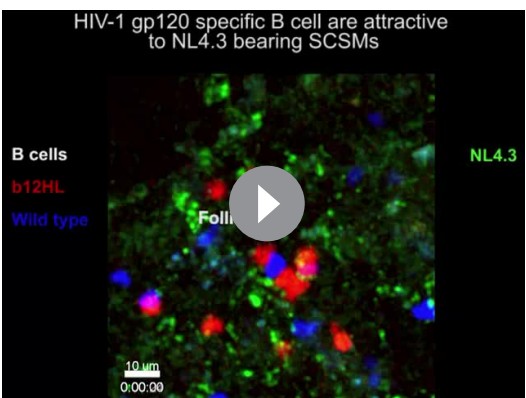

**Video 8.** b12 B cells attracted to, and in contact with NL4.3-GFP bearing SCSMs, while wild type B cells had much less interest. TP-LSM live imaging shows behavior of b12 (red) and wild type B (white) cells in inguinal LN. First video shows new arriving B cells in inguinal at 3 hr after NL4.3-GFP (green) injection. FDCs network revealed by injected CD21/35 (blue) antibodies 30 min before imaging. Images were acquired from 40 min after B cell transfer. HEV indicated base on transferred B cell attachment. B cell follicle (white circle) indicated by FDC network spreading. Sinus floor indicated by NL4.3-GFP accumulations in SCSMs. LN capsule was delineated by second harmonic signals (blue). Sinus indicated in between sinus floor and capsule. Second video shows behavior of B cells were adoptively transferred 1 day before the NL4.3-GFP injection. Images were acquired from 1 hr after NL4.3-GFP injection. Tracks of b12 (red) and wild type B cells (blue) were visualized with dragon tail which tracked path of each cells past 20 time points. SCSMs were delineated by NL4.3-GFP accumulations. Time counter is hour:minute:second.

https://elifesciences.org/articles/47776#video8

**Video 9.** b12 B cells and wild type B cells actively probe NL4.3-GFP bearing SCSMs but b12 B cells show longer contact duration and NL4.3-GFP bearing uropods. TP-LSM live imaging shows behavior of B cells were adoptively transferred 1 day before the NL4.3-GFP injection. Images were acquired from 3 hr after NL4.3-GFP injection. SCSMs were delineated by NL4.3-GFP accumulations. In first video b12 B cell (red) were highlighted with square box to indicate their long contact duration, while wild type B cells (blue) show much less interest. b12 B cell which acquired and accumulated NL4.3-GFP in its uropod is highlighted with white circle. The second video shows a SCSM bearing NL4.3-GFP to which a b12 and wild type B cells interact. Wild type B cell and SCSM contact was indicated with blue circle. The third portion of the video shows a b12 b cell actively probe NL4.3-GFP on SCSM with its filopodia. Final video shows a b12 B cell extracting NL4.3-GFP and pulling away with the VLP localized on its uropod. Time counter is hour:minute:second.

https://elifesciences.org/articles/47776#video9

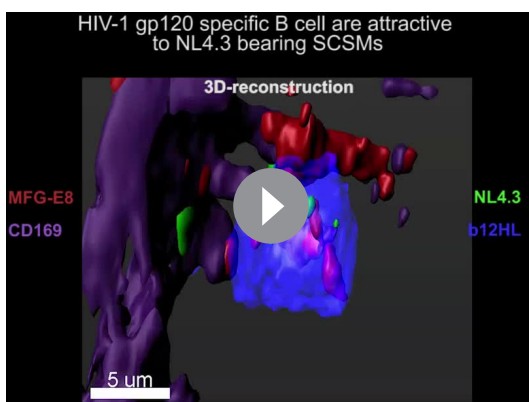

**Video 10.** b12 B cells actively extracting NL4.3-GFP with their filopodia by directly probing the SCSM MFG-E8+ compartment. Animation shows 3D-reconstructed images of b12 B cells (blue) associated with NL4.3-GFP (green) bearing SCSMs. b12 B cells were adoptively transferred 1 day before the NL4.3-GFP injection. Thick section from inguinal LN collected 3 hr after VLPs injection was stained with CD169 (purple) and MFG-E8 (red). Arrowhead indicates tight contact between b12 B cell and MFG-E8+ compartment which contains NL4.3-GFP. b12 B cell actively starched filopodia toward MFG-E8+ compartment and accumulated NL4.3-GFP in its cytoplasm.

https://elifesciences.org/articles/47776#video10

## Cells

Lymph node cells were isolated by following procedure. Inguinal LNs were carefully collected without fat tissue and gently teased apart with micro-forceps into RPMI 1640 media containing 2 mM L-glutamine, antibiotics (100 IU/ml penicillin, 100 µg/ml streptomycin), 1 mM sodium pyruvate, and 50 µM 2-mercaptoethanol, pH 7.2. The tissue was then digested with Liberase Blendzyme 2 (0.2 mg/ml, Roche Applied Science) and DNase I (20 µg/ml) for 30 min at 37°C, while rocking vigorously. Proteases were then inactivated with 10% fetal bovine serum and 2 mM EDTA and the cell disaggregated by passing them through a 40 µm nylon sieve (BD Bioscience). Single cells were then washed with 1% BSA/PBS and blocked with anti-Fcγ receptor (BD Biosciences). Splenic B cells were isolated by negative depletion using biotinylated antibodies to CD4, CD8, Gr-1 (Ly- 6C and Ly 6G), CD11b, and CD11c and Dynabeads M-280 Streptavidin (Thermo Fisher Scientific). The B cell purity was greater than 95%. When needed B cells were cultured in RPMI 1640 containing 10% fetal calf serum (FCS, Gibco), 2 mM L-glutamine, antibiotics (100 IU/mL penicillin and 100 µg/mL streptomycin), 1 mM sodium pyruvate, and 50 µM 2-mercaptoethanol. HEK293T cells were cultured in complete Dulbecco's modified Eagle's medium (DMEM) containing 4.5 g/L D-glucose, 4 mM L-glutamine, 3.7 g/L sodium bicarbonate, 10% tetracycline free FBS, 1 mM sodium pyruvate, and 1% penicillin streptomycin. Human peripheral blood mononuclear cell (PBMC) was purified by using a density gradient centrifugation with Ficoll (Ficoll-Paque, Miltenyi Biotec).

## Reagents

Reagents were purchased from indicated companies or program; αv blocking antibody (αCD51) (RMV-7, BioLegend); Mouse MFG-E8 Antibody (P21956), Human MFG-E8 Antibody (Q08431), Recombinant Human MFG-E8 Protein (Q08431), Recombinant Mouse MFG-E8 Protein (P21956), Recombinant Human EDIL-3 Protein (6046-ED-050), Recombinant Human Integrin alpha V beta 3 Protein (2308-VN, R and D Systems); Purified Rat Anti-Mouse Follicular Dendritic Cell (FDC-M1, BD Biosciences); Anti-HIV-1 gp120 Monoclonal (VRC01) (NIH AIDS Reagent Program); Pertussis toxin (PTx) (Millipore Sigma); Cyclo(-RGDyK) (AnaSpec); 1,2-dioleoyl-sn-glycero-3-phosphocholine (DOPC) (850375), 1-oleoyl-2-{6-((7-nitro-2–1,3-benzoxadiazol-4-yl)amino) hexanoyl}-sn-glycero-3-phosphocholine (NBD-PC) (810132), 1,2-dioleoyl-sn-glycero-3-phospho-L-serine (DOPS) (840035, Avanti Polar Lipids). Immune complexes (IC) were generated by mixing 4-Hydroxy-3-nitrophenylacetyl hapten-Phycoerythrin (NP-PE) (Biosearch Technologies) and anti-PE antibody (PE001, BioLegend). Briefly, NP-PE was incubated with anti-PE antibody at room temperature for 30 min using a 1:2 (NP-PE/antibody) ratio (weight:weight). Recombinant MFG-E8 was conjugated to fluorescent (Alexa Fluor 488, 594, or 647) with the Microscale Protein Labeling Kit (Thermo Fisher Scientific). Antibodies against to CD169 (3D6.112, BioLegend) were conjugated to Alexa Fluor 594 with the Antibody Labeling Kits (Thermo Fisher Scientific). Labeling reactions, conjugates purification, and determination of degree of labeling were performed following the company manuals.

## Viral-like particles preparation

Fluorescent HIV-1 VLPs (NL4.3-GFP) was produced by transfecting HEK293T cells with pCMV-NL4.3 Gag EGFP (*Guzzo et al., 2017*), which was kindly provide Walter Mothes (Yale University). NL4.3-mCherry was produced by transfecting HEK293T cells with NL4.3 Gag mCherry, which generated by switching the fluorescence protein from NL4.3 gag EGFP. Envelope deficient NL4.3-GFP VLP was produced by transfecting HEK293T cells with HIV-1 NL4-3 Gag-iGFP ΔEnv (12455, NIH AIDS Reagent Program). MLV-RFP VLPs were produced by triple transfection using pSV-A-MLV-env (1065, NIH AIDS Reagent Program), pSV-Ψ-MLV-env⁻ (3422, NIH AIDS Reagent Program) and MLV Gag-RFP (1814, Addgene). Briefly, HEK293T cells were transfected using TransIT-293 Transfection Reagent (Mirus Bio LLC). Eighty per cent confluent HEK293T in six well plates (2.5 ml/well) were transfected by adding dropwise to each well 250 µl containing 2.5 µg of plasmid diluted in Opti-MEM with a 1:4 (DNA/reagent) dilution of TransIT-293 Transfection Reagent. The media was harvested 72 hr later and centrifuged at $500 \times g$ for 10 min at 4°C. The supernatants were collected and mixed with Lenti-X Concentrator (Takara Bio USA, Inc) at a 1:3 ratio concentrator to supernatant. The mixture was placed at 4°C overnight and centrifuged the following day at $1500 \times g$ for 45 min. The pelleted VLPs were suspended in PBS and frozen in aliquots. HIV-1 p24 concentrations were measured with HIV-1 p24 in vitro SimpleStep ELISA (Enzyme-Linked Immunosorbent Assay) kit (Abcam). The numbers of HIV-1 VLPs in the final concentrates were counted as described below.

## Flowcytometry and VLP measurement

LIVE/DEAD Fixable Aqua Dead Cell Stain Kit (Molecular Probes) was used in all experiments to exclude dead cells. Single cells were re-suspended in PBS, 2% FBS, and stained with fluorochrome-conjugated antibodies against Gr-1 (RB6-8C5, Thermo Fisher Scientific), anti-F4/80 (BM8, Thermo Fisher Scientific), anti-CD11b (M1/70, Thermo Fisher Scientific), anti-CD11c (HL3, BD Bioscience), CD169 (3D6.112, BioLegend), and anti-B220 (RA3-6B2, BD Biosciences), anti-CD4 (clone RM4-5, BD Pharmingen), and anti-mouse MFG-E8 Antibody (R and D Systems). Human PBMC was analyzed with antibodies; anti-αvβ3 integrin (23C6), anti-CD14 (63D3), anti-CD4 (RPA-T4), anti-CD8 (RPA-T8), and anti-CD20 (2H7, BioLegend). Data acquisition was done on FACSCanto II flow cytometer (BD Biosciences) and analyzed with FlowJo software (Treestar). For NL4.3-GFP binding experiment suspended lymph node cells were incubated with NL4.3-GFP VLPs in 1% BSA/PBS at 4°C for 30 min. Binding mixtures were washed and re-suspended with 1% BSA/PBS. SCSMs were distinguished by antibody staining for surface markers. VLP binding was determined by strength of GFP signals. Fluorescent VLPs (NL4.3-GFP) were directly counted and measured by FACSCanto II flow cytometer. Flow cytometer which equipped with forward scatter (FSC) detector (Photodiode with 488/10 BP) and side scatter (SSC) detector (photomultiplier tube (PMT) with 488/10 BP) was tuned detection voltages up to 450 for FSC and SSC. Window extension and FSC Area Scaling was set to 8.0 and 1.0. NL4.3-GFP VLPs were distinguished from noise and non-VLP particles by GFP signals. GFP positive particles were confirmed as gp120 positive VLPs by anti-HIV-1 gp120 antibody (VRC01). For recombinant protein binding experiment to VLPs, $1 \times 10^4$ VLPs were suspended in 20 µl of 1% BSA/PBS or $1 \times$ HBSS (contains; 1 mM $Ca^{2+}$, 2 mM $Mg^{2+}$, 1 mM $Mn^{2+}$ and 0.5% fatty acid free BSA) buffer and incubated with recombinant proteins; EDIL-3 (5 µg/ml), MFG-E8 (5 µg/ml), fluorescent recombinant MFG-E8 (5 µg/ml) or integrin αvβ3 (5 µg/ml) on ice for 30 min. Non-label recombinant MFG-E8 and integrin αvβ3 was detected by antibody staining. HIV-1 gp120 antibody (VRC01) was stained with same protocol as above. Binding inhibition assay with liposomes (25 µg/ml) was performed with same protocol.

## Fluorescent liposomes

Fluorescent liposomes were generated with phospholipid mixtures by the following procedure (*Murakami et al., 2014*). Mixture compositions are; NBD PC liposome (99:1 = DOPS:NBD-PC) and NBD PS liposome (50:49:1 = DOPS:DOPC:NBD-PC). Large multilamellar vesicles were formed by swirling in degassed PBS (pH 7.4). Unilamellar vesicles were prepared by extrusion through filters with 100 nm pore size using Avanti Mini-Extruder (Avanti Polar Lipids). Sonicated mixtures were downsized using a small-scale extruder (Avestin Europe GmbH).

## Thick section immunohistochemistry and confocal microscopy

Immunohistochemistry was performed using a modified method of a previously published protocol (*Park et al., 2015*; *Park et al., 2018*). Briefly, freshly isolated LNs or spleens were fixed in newly prepared 4% paraformaldehyde (Electron Microscopy Science) overnight at 4°C on an agitation stage. Spleens or LNs were embedded in 4% low melting agarose (Thermo Fisher Scientific) in PBS and sectioned with a vibratome (Leica VT-1000 S) at a 30 µm thickness. Thick sections were blocked in PBS containing 10% fetal calf serum, 1 mg/ml anti-Fcγ receptor (BD Biosciences), and 0.1% Triton X-100 (Sigma) for 30 min at room temperature. Sections were stained overnight at 4°C on an agitation stage with the following antibodies: anti-B220 (RA3-6B2, BD Biosciences), anti-CD4 (RM4-5, BD Biosciences), anti-CD11b (M1/70, BioLegend), anti-CD169 (3D6.112, BioLegend), anti-CD21/35 (7E9, BioLegend), anti-CD35 (8C12, BD Biosciences) and anti-mouse MFG-E8 Antibody (R and D Systems) or with labeled MFG-E8. Stained thick sections were microscopically analyzed using a Leica SP8 confocal microscope equipped with an HC PL APO CS2 40× (NA, 1.30) oil objective (Leica Microsystem, Inc) and images were processed with Leica LAS AF software (Leica Microsystem, Inc) and Imaris software v.9.0.1 64× (Bitplane AG). The intensities of fluorescent signals in regions of interests (ROI) were measured by LSA AF Lite software (Leica Microsystem).

## Intravital two-photon laser scanning microscopy (TP-LSM)

Inguinal LNs were prepared for intravital microscopy as described (*Park et al., 2018*). Cell populations were labeled for 10 min at 37°C with 2.5-5 µM red cell tracker CMTMR (Thermo Fisher Scientific) or 2 µM of eFluor450 (Thermo Fisher Scientific). 5–10 million labeled cells of each population in 200 µl of PBS was adoptively transferred by tail vein injection into recipient mice. After anesthesia the skin and fatty tissue over inguinal LN were removed. The mouse was placed in a pre-warmed cover glass chamber slide (Nalgene, Nunc). The chamber slide was then placed into the temperature control chamber on the microscope. The temperature of air was monitored and maintained at $37.0 \pm 0.5°C$. Inguinal LN was intravitally imaged from the capsule over a range of depths (10–220 µm). All two-photon imaging was performed with a Leica SP8 inverted five channels confocal microscope (Leica Microsystems) equipped with 25 × water dipping objective, 0.95 NA (immersion medium used distilled water). Two-photon excitation was provided by a Mai Tai Ti:Sapphire laser (Spectra Physics) with a 10 W pump, tuned wavelength ranges from 820 to 920 nm. Emitted fluorescence was collected using a four channel non-descanned detector. Wavelength separation was through a dichroic mirror at 560 nm and then separated again through a dichroic mirror at 495 nm followed by 525/50 emission filter for GFP or Alexa Fluor 488 (Thermo Fisher Scientific); and the eFluor450 (Thermo Fisher Scientific) or second harmonic signal was collected by 460/50 nm emission filter; a dichroic mirror at 650 nm followed by 610/60 nm emission filter for CMTMR, PE or Alexa Fluor 594; and the Evans blue, Allophycocyanin (APC) or Alexa Fluor 647 signal was collected by 680/50 nm emission filter. For four-dimensional analysis of cell behavior, stacks of various numbers of section (z step = 2, 3, 4, 5 or 6 µm) were acquired every 5–30 s to provide an imaging volume of 20–120 µm in depth. Sequences of image stacks were transformed into volume-rendered four-dimensional videos using Imaris software v.9.0.1 64× (Oxford Instruments plc), and the tracks analysis was used for semi-automated tracking of cell motility in three dimensions by using the following parameters: autoregressive motion algorithm, estimated diameter 10 µm, background subtraction true, maximum distance 20 µm, and maximum gap size 3. Tracks acquired that could be tracked for at least 20% of total imaging duration were used for analysis. Some tracks were manually examined and verified. Calculations of the cell motility parameters (speed, track length, displacement, straightness and speed variability) were performed using the Imaris software v.9.0.1 64× (Oxford Instruments plc). Statistical analysis was performed using Prism software (GraphPad Software). 3D-reconstructions from original images from TP-LSM were generated by the surfaces function of the Surpass view in Imaris software v.9.0.1 64× (Oxford Instruments plc), performed with semi-automated creation wizard. The intensities of fluorescent signals in regions of interests (ROI) were measured by LSA AF Lite software (Leica Microsystem). Annotations on videos and video editing were performed using Adobe Premiere Pro CS3 (Adobe Systems Incorporated). Video files were converted to MPEG4 format with ImToo Video Converter Ultimate (v. 7.8.19) (ImToo Software Studio).

### Visualization of microanatomy in LN for intravital imaging

Microanatomy of LN delineated by direct antibody injection before live imaging. To outline blood vessels 50 µl of Evans Blue solution (0.5 µg/ ml in PBS) was injected into orbital or tail vein prior to imaging. To visualize SCSMs in the subcapsular sinuses, purified rat anti-mouse CD169 (3D6.112, BioLegend) was conjugated with Alexa Fluor 647. 0.5 µg of Alex Fluor 647 conjugated CD169 antibody in 50 µl of PBS was injected into tail base 1 hr prior to imaging. To visualize FDCs in the B cell follicles, 5 µg of PE conjugated rat anti-mouse CD21/35 (7E9, BioLegend) or anti-CD35 (8C12, BD Biosciences) was injected into tail base 1 hr prior to imaging. Intravital imaging was performed as described above.

### STED microscopy

STED microscope imaging for VLPs using modified protocol from previous report (*Chojnacki et al., 2012*). Labeled recombinant MFG-E8-Alexa Fluor 594 (250 ng) was incubated with NL4.3-GFP VLPs ($0.1 \times 10^6$ of GFP positive particles) at 4°C for 30 min. Ten micro liters of MFG-E8/NL4.3-GFP mixture mixed with 40 µl of mounting solution (ProLong Gold, Life Technologies). Ten micro liters of final mixture was mounted on a slide covered with #1 coverslip. All STED imaging was performed with a Leica DMI6000 SP8X CW gated STED system with a 775 nm depletion laser (STED dichroic slider) and an HC PL APO CS2 100× (NA, 1.40) oil objective. Pixel size was set to 19 nm. Images were acquired with Leica LAS AF software (Leica Microsystem, Inc) and processed Imaris software v.9.0.1 64× (Oxford Instruments plc). Raw images were filtered with Gaussian filter which is standard function of Imaris software. The intensities of fluorescent signals in regions of interests (ROI) were measured by LSA AF Lite software (Leica Microsystem).

### In vitro imaging of HEK293T cells and VLPs

HEK293T cells were transfected using TransIT-293 Transfection Reagent (Mirus Bio LLC). Same transfection protocol with VLP preparation was applied for imaging experiments. Transfected HEK293T cells were imaged with a Leica SP8 inverted five channels confocal microscope (Leica Microsystems) equipped with 40 × oil objective, 0.95 NA (immersion medium used distilled water). The temperature of air (5% $CO_2$) was maintained at 37.0 ± 0.5°C. VLPs were detected by GFP signals. MFG-E8 was visualized by antibody (1 µg/ml) addition into culture media. Fluorescent MFG-E8 (250 ng/ml) which was added into culture media visualized by its own signal. For NL4.3-GFP/MFG-E8 capture experiment, eFluor450 (Thermo Fisher Scientific) labeled b12HL B cells were directly added into culture of HEK293T cells and imaged. Images were acquired with Leica LAS AF software (Leica Microsystem, Inc) and processed Imaris software v.9.0.1 64× (Oxford Instruments plc). Raw images were filtered with Gaussian filter which is standard function of Imaris software.

### Quantifications and statistical analyses

All experiments were performed at least three times. Represent images were placed in figures. Primary image data which was analyzed and calculated by Leica LAS AF software (Leica Microsystem, Inc) or Imaris software v.9.0.1 64× (Oxford Instruments plc) was acquired and processed with Microsoft Excel software. Error bars with ± SEM, and p values were calculated with unpaired t-test in GraphPad Prism 9.01 (GraphPad software). $p < 0.05$ was considered significantly different.

## Acknowledgements

The authors are grateful to Dr. David Nemazee for providing the b12 KI mice, Dr. Mark Udey for providing *Mfg-e8*$^{-/-}$ mice, Dr. John Coligan for providing the PS and PC liposomes, Dr. Claudia Cicala for critical reading of the manuscript and helpful discussions, and Dr. Anthony Fauci for his continued support. This research was supported by the intramural program of the National Institutes of Allergy and Infectious Diseases.

## Additional information

### Funding

| Funder | Author |
| --- | --- |
| National Institutes of Health | Chung Park<br>John H Kehrl |

No external funding was received for this work.

### Author contributions

Chung Park, Conceptualization, Data curation, Formal analysis, Supervision, Investigation, Visualization, Methodology; John H Kehrl, Conceptualization, Supervision, Project administration

### Author ORCIDs

Chung Park (iD) https://orcid.org/0000-0002-7819-5333
John H Kehrl (iD) https://orcid.org/0000-0002-6526-159X

### Ethics

Human subjects: Human peripheral blood mononuclear cells (PBMCs) were collected from healthy donors through a NIH Department of Transfusion Medicine protocol that was approved by the Institutional Review Board of the National Institute of Allergy and Infectious Diseases (NIAID), National Institutes of Health.
Animal experimentation: The NIAID Animal Care and Use Committee (ACUC) at the National Institutes of Health approved all the animal experiments and protocols used in the study, under protocol LIR-15E.

### Decision letter and Author response

Decision letter https://doi.org/10.7554/eLife.47776.sa1
Author response https://doi.org/10.7554/eLife.47776.sa2

## Additional files

### Supplementary files

• Transparent reporting form

### Data availability

All data generated or analysed during this study are included in the manuscript and supporting files.

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
