## [Decision Letter]

Thank you for submitting your article "Αn Integrin/MFG-E8 shuttle loads HIV-1 viral like particles onto follicular dendritic cells" for consideration by *eLife*. Your article has been reviewed by three peer reviewers, and the evaluation has been overseen by a Reviewing Editor and Päivi Ojala as the Senior Editor. The following individual involved in review of your submission has agreed to reveal their identity: Ziv Shulman (Reviewer #2).

The reviewers have discussed the reviews with one another and the Reviewing Editor has drafted this decision to help you prepare a revised submission.

All three reviewers found the work exciting. In particular, the notion that integrins are involved in the trapping and presentation of VLPs was noted as an important and novel finding. However, as you will see from their comments attached below, they raised several issues that will need attention in the revised version of the manuscript. In particular, and most importantly, reviewers were unconvinced about the claim on "FDC chemotaxis". All three found these observations interesting but felt that the tone of the claims would need to be tuned down or better substantiated experimentally. A possibility that must be considered in addressing this is that these observations could be the result of "B cell pushing FDC dendrites". Using CD35 stainings will be an important addition. While it appears to be a consensus about this being the main major point, we will encourage you to carefully go through the reviewer's comments to address as much as you can the other points raised. This will help to improve the manuscript as a whole.

Reviewer #1:

This manuscript by Park and Kehrl reports on an analysis of pathways governing the capture of viral particles, particularly HIV-mimetic VLPs, in the lymph nodes, and the mechanisms governing transfer of viral particles to antigen-specific B cells and the FDC network in follicles. They find that HIV virus-like particles and murine leukemia virus VLPs are captured by a subset of subcapsular sinus macrophages with enriched surface localization of the innate immune protein MFG-E8. They present evidence that MFG-E8 binds to phosphatidyl serine exposed on the VLP membranes, bridging the particles to αvβ3 integrins expressed by the macrophages, and facilitating transfer to the FDC network in underlying follicles as well as antigen-specific B cells. The work is detailed and nicely encompasses in vivo experiments supplemented by more controlled in vitro analysis of binding interactions with cells/proteins and purified viral particles. The visualization of a dynamic remodelling of the FDC network to bring FDCs in contact with the overlying macrophage layer shortly following VLP administration is a striking finding. Overall, these findings provide a number of interesting new insights into how viral particles may be recognized and trafficked in lymphoid tissues.

There are some points that should be discussed and/or addressed through experimental clarification:

1) It is unclear how the authors view their findings fit in with the studies from the Sewald et al., 2015 study cited in the manuscript, where CD169 was shown to be important for HIV VLP capture; blockade or CD169 KO animals showed greatly reduced HIV VLP capture. In contrast, here the authors show greatly reduced VLP capture by blocking αv integrins (Figure 4A). How does one reconcile these distinct observations?

2) The authors show in Figure 2D data indicating that substantial αvβ3 binding to the HIV VLPs even in the absence of MFG-E8 (this is noted as "low level" binding in the text but it is more than a log over background). This seems to be consistent with the observations in vivo in Mfg-e8^-/-^ mice where VLP capture does not appear to be disrupted. Is the interpretation that MFG-E8 is really mainly important in the transfer to FDCs/B cells and not initial capture?

3) Some of the flow cytometry binding experiments (e.g., Figure 1G, Figure 2D, Figure 3D/E) are presented as single sample examples, but mean binding results from repeat experiments should be shown, particularly for the human PBMC binding experiments in Figure 3. Particularly in light of the authors' comments about variability in donor monocyte binding, which is not shown.

4) The closing results statement, that antigen-specific B cells were attracted to and in contact with VLP-bearing SCSMs is not clearly supported by the quantification data in Figure 7E. It's not clear that speed, displacement, or track length are indicative of interactions with the SCSMs. (What is the definition of "straightness"? Is this a permutation of persistence length?). It is also unclear why the antigen-specific cells would be attracted toward the SCSMs but not the wild type cells, if this behavior is driven by some type of chemoattractant before specific antigen encounter.

5) The study focuses on the trafficking behavior of free virions but cell-associated viral dissemination is thought to be important during real infection. What is the evidence for a role of free virus in dissemination during the early stages of live infection? This would be valuable to discuss either in the Introduction to motivate the study or the Discussion to provide context for the results.

6) In the in vitro analysis of binding of VLPs to immune cells, it was noted that VLPs did not bind to T cells or B cells, but the Discussion suggests that MFG-E8-bound VLPs could be engaged by αvβ3 integrins expressed by B cells. Do B cells express αvβ3 or not? This discrepancy should be addressed.

Reviewer #2:

The study by Park and Kehrl examined how virus-like particles are captured by subcapsular macrophages in lymph nodes and deliver them to follicular dendritic cells. The authors inject NL4.3-GFP and then track its localization in vivo. They find an accumulation of these VLP in subscapular macrophages and at a later time point, they detect this protein on FDC network. Similar results were obtained with MLV. Using the FDC-M1 antibody, the authors detected a signal on SCSM which indicates that MFG-E8 is expressed on these cells and this signal localized with the VLP particles by microscopy assay. The authors provide additional evidence for VLP binding on MFG-E8-expressing cells by flow cytometry. A similar phenomenon was observed when VLP were injected intravenously and marginal zone macrophages were examined. The authors show in vitro experiments that prove the binding of VLPs to MFG-E8 and αvβ3integrins. These complexes were also detected in the histology section made from mouse lymph nodes and observed on human monocytes. in vivo, the uptake of NL4.3-GFP by monocytes and FDC was inhibited by αvβ3 blockage. The authors show that MFG-E3 is required for uptake of VLP by FDCs but not by SCSM. The authors demonstrate that EDIL-3 can compensate for MFG-E3 binding. VLP binding to SCSM overlapped with the immune complex bindings. In response to VLP, the FDC network interacts with SCSM in Gai dependent manner. Antigen-specific B cells interact with SCSM bearing VLP and capture antigen from these cells.

The study shows nicely the location of VLP particles and the mechanism that control its dissemination to the FDC network. The study used imaging techniques that address the question in an elegant way. One major issue is whether the NL4.3-GFP represent a virus without GFP and whether the local or intravenous injection of many VLP represents a physiological condition. Nonetheless, the model shed light on unknown mechanism that most likely plays a role in viral infection. Furthermore, the authors examine FDCs using an antibody which is not very specific for these cells. In order to make solid claims, the authors should use the anti-CD35 antibody at least in some of the experiments.

Overall, in my opinion, with some additional improvements, the study would be a nice fit to *eLife*.

1) The interaction between the FDC and SCSM makes a lot of sense and this is a very important point in the study. In Figure 1D, the authors should use a more specific antibody for the FDC. Anti-CD35-biotin is available and should be simple to stain with, and help to precisely prove their point about interaction between the SCSM and FDCs. It would also be nice to show CD35 in inflamed lymph node as well.

2) The video that shows movement of the FDC network is stunning. However, it is very unlikely that these cells can move at such a speed. These are not lymphocytes or immune cells. To me, it seems that movement of B cells, in a Gai-dependent manner, push the network and therefore it seems to move. Without additional proof, the authors can only state that network comes closer to the lymph node cortex, rather than claiming that FDC actually moves.

Reviewer #3:

While studying capture HIV and MLV virus particles by SCS macrophages authors made the probably serendipitous (using the anti-MFG-E8 Ab FDC-M1 to stain FDC), but nevertheless very interesting observation that enveloped viral like particles co-localize with MFG-E8, and following up with experiment that indicate a potential role for MFG-E8 on VLP capture by SCS macrophages and transfer to FDCs. Using an array of in vivo and sophisticated in vitro assays they convincingly demonstrate the interaction between PS on VLP and MFG-E8 and that these complexes can bind to αv integrins. The strongest aspects of the study are to me the demonstration of MPG-E8-VLP co-localization in vivo and the pronounced reduction of SCS deposition of VLPs upon αv-combination blockade with antibodies and RGD peptide. However, the data showing the chemoattractant driven shift of FDC network toward the SCS macrophages and extraction of VLPs by B cells are not convincing.

My suggestion is that the authors tighten up the interesting in vivo demonstration of the αv and MFG-E8-dependent viral capture mechanism, and remove the studies on FDC chemotaxis towards the SCS and cognate B cell capture of VLPs.

1) One obvious question is why the authors decided to study HIV and not a relevant murine pathogen? They insert some nice human data on monocytes expressing αvβ3, but overall the disconnect between a strictly human pathogen and mouse hosts are a concern. While some mechanistic studies are of course only possible in the mouse system, additional human validation would be helpful to make this a strong HIV paper.

It was a bit surprising that authors did not followed up on their initial MLV observations. Did they consider species-specific differences for MFG-E8 functions (one vs. two RGD motif binding domains)? Is the αv expression pattern on mouse vs. human immune cells comparable?

2) Immuno-staining for αv integrin on LN section isn't fully convincing (Figure 3A). Do the authors have positive or negative controls to validate their staining? αv integrins are widely expressed, e.g. on inflamed epithelium, on fibroblasts (αvβ6), or on dendritic cells (αvβ3 and αvβ5, αvβ8).

Maybe β3 or β5 (which only pair with αv) are better targets for fluorescence imaging?

Based on the data shown I am not convinced that the conclusion of co-localization of αv with MFG-E8 and HIV VLPs in vivo is justified (even though it appears intuitive, given prior reports on MFG-E8-αvβ3/5 integrin-interactions).

3) A conceptual problem with the in vivo imaging studies is that reagents (e.g. fluorescent immune complexes and antibodies against CD169 and CD21/35) that may go through the same Ag transport pathway through tissues as VLPs are used to delineate the cells involved in that pathway. How can it be ascertained that these reagents do not also partially stain adjacent cells.

4) There is not effort to confirm an effect of PTX on the motility of the FDC network. PTX will predictably affect B cell motility and thereby also impair any B cell-dependent FDC loading mechanism.

Looking at the videos that are claimed to illustrate FDC chemotaxis, I can't help but suspect that the FDC movement is caused by phototoxic effects, most obviously at time 00:20 min:sec of Video 5.

---

## [Author Response]

Reviewer #1:This manuscript by Park and Kehrl reports on an analysis of pathways governing the capture of viral particles, particularly HIV-mimetic VLPs, in the lymph nodes, and the mechanisms governing transfer of viral particles to antigen-specific B cells and the FDC network in follicles. They find that HIV virus-like particles and murine leukemia virus VLPs are captured by a subset of subcapsular sinus macrophages with enriched surface localization of the innate immune protein MFG-E8. They present evidence that MFG-E8 binds to phosphatidyl serine exposed on the VLP membranes, bridging the particles to αvβ3 integrins expressed by the macrophages, and facilitating transfer to the FDC network in underlying follicles as well as antigen-specific B cells. The work is detailed and nicely encompasses in vivo experiments supplemented by more controlled in vitro analysis of binding interactions with cells/proteins and purified viral particles. The visualization of a dynamic remodeling of the FDC network to bring FDCs in contact with the overlying macrophage layer shortly following VLP administration is a striking finding. Overall, these findings provide a number of interesting new insights into how viral particles may be recognized and trafficked in lymphoid tissues.There are some points that should be discussed and/or addressed through experimental clarification:1) It is unclear how the authors view their findings fit in with the studies from the Sewald et al., 2015 study cited in the manuscript, where CD169 was shown to be important for HIV VLP capture; blockade or CD169 KO animals showed greatly reduced HIV VLP capture. In contrast, here the authors show greatly reduced VLP capture by blocking α-v integrins (Figure 4A). How does one reconcile these distinct observations?

Our study indicates that SCSMs can use variety of strategies to capture VLPs and that CD169 and MFG-E8 may have overlapping roles perhaps functioning at different stages of the capture process. We added this in the Discussion.

2) The authors show in Figure 2D data indicating that substantial αvβ3 binding to the HIV VLPs even in the absence of MFG-E8 (this is noted as "low level" binding in the text but it is more than a log over background). This seems to be consistent with the observations in vivo in Mfg-e8^-/-^ mice where VLP capture does not appear to be disrupted. Is the interpretation that MFG-E8 is really mainly important in the transfer to FDCs/B cells and not initial capture?

It is difficult to determine whether MFG-E8 functionally promotes HIV-1 capture by SCSMs using the MFG-E8 deficient animals. As the reviewer noted it is likely that some MFG-E8 is already associated with VLPs prep, which can promote αv integrin mediated SCSM capture. Also, in the absence of MFG-E8, Edil-3 may compensate for its loss. Lastly the decreased release of the captured VLPs by the MFG-E8^-/-^ SCSM gives the impression that the MFG-E8^-/-^ SCSM can efficiently capture the VLPs. While we suspect that MFG-E8 promotes SCSM uptake, our data only allows us to conclude that integrin blockade inhibits SCSM VLP uptake and that MFG-E8 functions in the transfer of the VLPs to the FDC network. We have clarified our conclusions in the revised manuscript.

3) Some of the flow cytometry binding experiments (e.g., Figure 1G, Figure 2D, Figure 3D/E) are presented as single sample examples, but mean binding results from repeat experiments should be shown, particularly for the human PBMC binding experiments in Figure 3. Particularly in light of the authors' comments about variability in donor monocyte binding, which is not shown.

We added data to support this comment into the supplementary figures.

4) The closing results statement, that antigen-specific B cells were attracted to and in contact with VLP-bearing SCSMs is not clearly supported by the quantification data in Figure 7E. It's not clear that speed, displacement, or track length are indicative of interactions with the SCSMs. (What is the definition of "straightness"? Is this a permutation of persistence length?). It is also unclear why the antigen-specific cells would be attracted toward the SCSMs but not the wild type cells, if this behavior is driven by some type of chemoattractant before specific antigen encounter.

We agree with the reviewer that we did not clearly explain our results and we have reworded and further explained them. Both the wild type and b12 B cells are attracted to and make contacts with VLP bearing SCSMs. Our tracking data examined the behavior patterns of all the B cell tracks from the intravital imaging space. Although WT and b12 B cell tracks did not have significantly different displacements in whole area of imaging field, the b12 B cells had a reduced mean speed, turned more frequently (moved less straight). When we focused on those tracks that interacted with the SCSMs, we found that the b12 B cell contacted the VLP bearing SCSMs for significantly longer durations than did the WT B cells (Figure 7F). We added the contact data and better explained the motility parameters in the main text.

5) The study focuses on the trafficking behavior of free virions but cell-associated viral dissemination is thought to be important during real infection. What is the evidence for a role of free virus in dissemination during the early stages of live infection? This would be valuable to discuss either in the Introduction to motivate the study or the Discussion to provide context for the results.

We are aware that their remains a controversy in the HIV field about the relative role of cell-free versus cell-associated virus in the spread of HIV. Yet it is well established that there are significant levels of cell free virus in plasma during acute and chronic infection. Surprisingly there is a paucity of information on the amount of free virus in the lymph. However, as much of the plasma virus is derived from tissue sources, it seems likely that it has transited through the lymph. Furthermore, as lymph nodes are linked in chains, the efferent lymph of primary lymph nodes serves as afferent lymph for the secondary nodes. Viral replication in the primary lymph node likely result in the transit of free-virus to the next node in the chain. Our study was directed at understanding how lymph and blood borne viral particles are delivered and accumulate upon FDCs. We believe our data is relevant not only for HIV infection, but also for the use of viral like particles as vaccine candidates. While the number of VLPs we locally inoculated (0.5 million) likely exceeds the numbers present in lymph some patients have plasma free virus levels in that range. Also arguing that cell free virus transits the lymph, large amounts of free virus accumulate on FDCs in patients. We have added some discussion relevant to this point.

6) In the in vitro analysis of binding of VLPs to immune cells, it was noted that VLPs did not bind to T cells or B cells, but the Discussion suggests that MFG-E8-bound VLPs could be engaged by αvβ3 integrins expressed by B cells. Do B cells express αvβ3 or not? This discrepancy should be addressed.

Murine germinal center B cells express αvβ3 integrins, while non-GC B cells do not. Human blood B cells lack αvβ3 integrins, but human lymphoblastoid B cell lines can express αvβ3 integrins. As far as we know human GC B cell expression has not been reported. In human blood only monocytes are likely to capture MFG-E8 bound virions. Interestingly, a recent study indicates that murine CD4 follicular helper T cells can also express αv (Schrock DC et al. Proc Natl Acad Sci U S A. 2019 Feb 15) providing another potential mechanism by which HIV viral particles can bind to CD4 T cells in lymph nodes. In our study the resting lymph node lymphocytes did not express αvβ3 integrins.

Reviewer #2:The study by Park and Kehrl examined how virus-like particles are captured by subcapsular macrophages in lymph nodes and deliver them to follicular dendritic cells. The authors inject NL4.3-GFP and then track its localization in vivo. They find an accumulation of these VLP in subscapular macrophages and at a later time point, they detect this protein on FDC network. Similar results were obtained with MLV. Using the FDC-M1 antibody, the authors detected a signal on SCSM which indicates that MFG-E8 is expressed on these cells and this signal localized with the VLP particles by microscopy assay. The authors provide additional evidence for VLP binding on MFG-E8-expressing cells by flow cytometry. A similar phenomenon was observed when VLP were injected intravenously and marginal zone macrophages were examined. The authors show in vitro experiments that prove the binding of VLPs to MFG-E8 and αvβ3 integrins. These complexes were also detected in the histology section made from mouse lymph nodes and observed on human monocytes. in vivo, the uptake of NL4.3-GFP by monocytes and FDC was inhibited by αvβ3 blockage. The authors show that MFG-E3 is required for uptake of VLP by FDCs but not by SCSM. The authors demonstrate that EDIL-3 can compensate for MFG-E3 binding. VLP binding to SCSM overlapped with the immune complex bindings. In response to VLP, the FDC network interacts with SCSM in Gai dependent manner. Antigen-specific B cells interact with SCSM bearing VLP and capture antigen from these cells.The study shows nicely the location of VLP particles and the mechanism that control its dissemination to the FDC network. The study used imaging techniques that address the question in an elegant way. One major issue is whether the NL4.3-GFP represent a virus without GFP and whether the local or intravenous injection of many VLP represents a physiological condition. Nonetheless, the model shed light on unknown mechanism that most likely plays a role in viral infection. Furthermore, the authors examine FDCs using an antibody which is not very specific for these cells. In order to make solid claims, the authors should use the anti-CD35 antibody at least in some of the experiments.Overall, in my opinion, with some additional improvements, the study would be a nice fit to eLife.1) The interaction between the FDC and SCSM makes a lot of sense and this is a very important point in the study. In Figure 1D, the authors should use a more specific antibody for the FDC. Anti-CD35-biotin is available and should be simple to stain with, and help to precisely prove their point about interaction between the SCSM and FDCs. It would also be nice to show CD35 in inflamed lymph node as well.

We added data for CD35 stain in Figure 4—figure supplement 3. In this study we already indicated that VLPs strongly initiated FDC abutment on SCSMs. In this new supplementary figure we indicated that CD35 visualized FDC abutments on SCSMs in VLP injected LN. As a follow up to this study we are trying to identify the signals that trigger SCSM/FDCs abutment and to uncover the mechanism of FDC movement.

2) The video that shows movement of the FDC network is stunning. However, it is very unlikely that these cells can move at such a speed. These are not lymphocytes or immune cells. To me, it seems that movement of B cells, in a Gai-dependent manner, push the network and therefore it seems to move. Without additional proof, the authors can only state that network comes closer to the lymph node cortex, rather than claiming that FDC actually moves.

We agree with the reviewer that we cannot state that the FDC network movement does not depend upon the movement of the B cells as they are intimately associated. Since pertussis toxin will inactivate Gα_i_ signaling in both B cells and FDCs, the loss of FDC/SMSM abutment could be dependent on either cell type or both. Assessing the FDC network in relation to the transferred B cells did suggest that the FDC network movement occurred independent of pressure from the B cells, yet this is by no means proof. By confirming the FDC/SCSM abutment by section imaging we can conclude that this phenomenon was not secondary to photo toxicity or secondary to surgically generated disturbance of the vasculature. Further studies will be needed to deactivate Gα_i_ signaling in the FDCs without simultaneously inhibiting B cell Gα_i_ signaling.

Reviewer #3:While studying capture HIV and MLV virus particles by SCS macrophages authors made the probably serendipitous (using the anti-MFG-E8 Ab FDC-M1 to stain FDC), but nevertheless very interesting observation that enveloped viral like particles co-localize with MFG-E8, and following up with experiment that indicate a potential role for MFG-E8 on VLP capture by SCS macrophages and transfer to FDCs. Using an array of in vivo and sophisticated in vitro assays they convincingly demonstrate the interaction between PS on VLP and MFG-E8 and that these complexes can bind to αv-integrins. The strongest aspects of the study are to me the demonstration of MPG-E8-VLP co-localization in vivo and the pronounced reduction of SCS deposition of VLPs upon αv-combination blockade with antibodies and RGD peptide. However, the data showing the chemoattractant driven shift of FDC network toward the SCS macrophages and extraction of VLPs by B cells are not convincing.My suggestion is that the authors tighten up the interesting in vivo demonstration of the αv and MFG-E8-dependent viral capture mechanism, and remove the studies on FDC chemotaxis towards the SCS and cognate B cell capture of VLPs.1) One obvious question is why the authors decided to study HIV and not a relevant murine pathogen? They insert some nice human data on monocytes expressing αvβ3, but overall the disconnect between a strictly human pathogen and mouse hosts are a concern. While some mechanistic studies are of course only possible in the mouse system, additional human validation would be helpful to make this a strong HIV paper.It was a bit surprising that authors did not followed up on their initial MLV observations. Did they consider species-specific differences for MFG-E8 functions (1 vs. two RGD motif binding domains)? Is the αv expression pattern on mouse vs. human immune cells comparable?

We certainly agreed with the reviewer that the mouse immune system cannot fully replicate the mechanisms underlying HIV dissemination in humans. But, the major focus of this study was how and when HIV viral particles becomes located on FDCs in lymphoid organ during the early stage of HIV infection. We chose to use HIV-1 VLPs and the mouse lymphatics and lymph node, as we can readily observe them using intravital microscopy. We initially compared MLV-VLPs and HIV-1 VLPs to determine whether there were any apparent differences in their SCSM uptake and their delivery to FDCs. Since we found no critical difference, it supported our usage HIV-1 VLPs in the mouse system. We did compare human and mouse MFG-E8 and found both could efficiently bound to HIV-1 VLPs and both bound αv integrins of either human or mouse origin. Here is our response to reviewer 1 who also asked about human versus murein αv expression- Murine germinal center B cells express αvβ3 integrins, while non-GC B cells do not. Human blood B cells lack αvβ3 integrins, but human lymphoblastoid B cell lines can express αvβ3 integrins. As far as we know human GC B cell expression has not been reported. In human blood only monocytes are likely to capture MFG-E8 bound virions. Interestingly, a recent study indicates that murine CD4 follicular helper T cells can also express αv (Schrock DC et al. Proc Natl Acad Sci U S A. 2019 Feb 15) providing another potential mechanism by which HIV viral particles can bind to CD4 T cells in lymph nodes. In our study the resting lymph node lymphocytes did express αvβ3 integrins.

2) Immuno-staining for αv integrin on LN section isn't fully convincing (Figure 3A). Do the authors have positive or negative controls to validate their staining?

We added Figure 3—figure supplement 1 for isotype control stain and Figure 4—figure supplement 1D for negative controls. And we also used an internal positive or negative control (VLP signal only, CD51 signal only) to verify CD51 staining signal (Figure 3—figure supplement 2 for isotype control stain).

αv integrins are widely expressed, e.g. on inflamed epithelium, on fibroblasts (αvβ6), or on dendritic cells (αvβ3 and αvβ5, αvβ8).Maybe β3 or β5 (which only pair with aV) are better targets for fluorescence imaging?

As the reviewer states αv is a partner for several β chains (β1, β3, β5, β6 and β8). Unfortunately, we could not find β3 and β5 antibodies that gave reliable staining with our methods, therefore we had to rely on αv antibodies. Since MFG-E8 binds αvβ3 and αvβ5, the MFG-E8 co-localization with αv antibodies is likely with those integrins.

Based on the data shown I am not convinced that the conclusion of co-localization of αv with MFG-E8 and HIV VLPs in vivo is justified (even though it appears intuitive, given prior reports on MFG-E8- αvβ3/5 integrin-interactions).

To provide more evidence that supports this issue, we added Figure 3—figure supplement 2. And we consider that the FACS data (Figure 2D) strongly supports the direct binding of αvβ3 integrins and MFG-E8 bound VLPs.

3) A conceptual problem with the in vivo imaging studies is that reagents (e.g. fluorescent immune complexes and antibodies against CD169 and CD21/35) that may go through the same Ag transport pathway through tissues as VLPs are used to delineate the cells involved in that pathway. How can it be ascertained that these reagents do not also partially stain adjacent cells.

We agree that this could be a concern. So, we verified our intravital imaging data with section staining and confocal microscopy, which does not require in vivo antibody transfer. All the data was consistent with the intravital in vivo observations.

4) There is not effort to confirm an effect of PTX on the motility of the FDC network. PTX will predictably affect B cell motility and thereby also impair any B cell-dependent FDC loading mechanism.

We agree that PTX treatment will affect all the cells in LN. However, in our study we have not observed any non-cognate B cells, which have bound VLPs. We readily observed direct transfer of the VLPs from the SCSMs to FDCs. Thus, we have concluded B cell mediated FDC loading is not the major mechanism by which the VLPs are transferred from SCSMs to FDCs and that the most likely mechanism by which PTX treatment disrupts VLP delivery is by disconnection FDC abutment on SDSM. Nevertheless, we have revised the manuscript and added several sentences to the Discussion tempering our conclusions.

“interfering with the SCSM/FDC abutment by pre-injecting pertussis toxin severely reduced the rapid VLP loading on the FDCs. The reduced SCSM/FDC abutment by pertussis toxin argues that heterotrimeric Gi protein-mediated signaling contributes to the FDC movement. The most intriguing possibility is a chemoattractant directed movement of the FDC network towards the SCSMs triggered by VLP uptake. However, since the local pertussis toxin injection will affect all the LN cells near the SCS, we cannot conclude that the FDC movement and severe reduction in FDC/VLP loading results solely from reducing the SCSM/FDC abutment.”

Looking at the videos that are claimed to illustrate FDC chemotaxis, I can't help but suspect that the FDC movement is caused by phototoxic effects, most obviously at time 00:20 min:sec of Video 5.

This is similar comment with major comment 2 of reviewer #2. We provide the same response:

We agree with the reviewer that we cannot state that the FDC network movement does not depend upon the movement of the B cells as they are intimately associated. Since pertussis toxin will inactivate Gα_i_ signaling in both B cells and FDCs, the loss of FDC/SCSM abutment could be dependent on either cell type or both. Assessing the FDC network in relation to the transferred B cells did suggest that the FDC network movement occurred independent of pressure from the B cells, yet this is by no means proof. By confirming the FDC/SCSM abutment by section imaging we can conclude that this phenomenon was not secondary to photo toxicity or secondary to surgically generated disturbance of the vasculature. Further studies will be needed to deactivate Gα_i_ signaling in the FDCs without simultaneous inhibiting B cell Gα_i_ signaling.